# A restricted dynamic surface self-reconstruction toward high-performance of direct seawater oxidation

Ling Zhou[1], Daying Guo [1] ✉, Lianhui Wu[1], Zhixi Guan[1], Chao Zou [1], Huile Jin[1], Guoyong Fang[1], Xi'an Chen [1] ✉ & Shun Wang [1] ✉

The development of highly efficient electrocatalysts for direct seawater splitting with bifunctionality for inhibiting anodic oxidation reconstruction and selective oxygen evolution reactions is a major challenge. Herein, we report a direct seawater oxidation electrocatalyst that achieves long-term stability for more than 1000 h at 600 mA/cm²@$\eta_{600}$ and high selectivity (Faraday efficiency of 100%). This catalyst revolves an amorphous molybdenum oxide layer constructed on the beaded-like cobalt oxide interface by atomic layer deposition technology. As demonstrated, a new restricted dynamic surface self-reconstruction mechanism is induced by the formation a stable reconstructed Co-Mo double hydroxide phase interface layer. The device assembled into a two-electrode flow cell for direct overall seawater electrolysis maintained at 1 A/cm²@1.93 V for 500 h with Faraday efficiency higher than 95%. Hydrogen generation rate reaches 419.4 mL/cm²/h, and the power consumption (4.62 KWh/m³ H₂) is lower than that of pure water (5.0 KWh/m³ H₂) at industrial current density.

Direct seawater electrolysis for hydrogen production is one of the effective ways to convert intermittent energy, such as solar energy, wind energy, and tidal energy, into chemical energy[1-4]. Nevertheless, the conduction of seawater splitting remains a serious challenge to ensure its efficiency, especially for the anodic oxygen evolution reaction (OER). The OER tends to suffer from sluggish kinetics due to its four-electron transfer process. Fortunately, the various catalysts including S-(Ni, Fe)OOH, NiMoN@NiFeN, etc., have been designed to accelerate the kinetics to reduce the overpotential of OER to some extent[5-8]. However, these used catalysts are prone to initiate the self-restructuring reaction between the catalysts and high-activity nascent intermediates (e.g., O*, HO*, and HOO*) under the OER high overpotential, thus forming hydroxyl metal oxides and further generate high-valent metal oxides[9-13]. These reconstructions caused the destruction of the catalyst structure, especially the high corrosion of seawater electrolyte, thus, eventually, most of the catalysts were seriously deactivated[14-16]. Interestingly, the surface of the catalyst is reconstructed to form hydroxyl metal oxides in the potential range, which is considered the "real catalyst" of OER[17-22]. For example, Wang et al. reported that the oxidation state of Co³⁺ at the octahedral site in Co₃O₄-based catalysts remained unchanged at the anodic potential, while Co²⁺ at the tetrahedral site was oxidized to form the CoOOH intermediate species, thus obtaining OER catalyst with high activity[23]. Lim et al. proposed a strategy of cationic oxidation-reduction to regulate the in situ leaching of OER electrocatalysts and realized the directional dynamic surface reconstruction of layered LiCoO₁.₈Cl₀.₂ in the OER process[24]. However, this continuous and disordered reconstruction process is fatal to stabilizing the structure of the catalyst[16,25-27]. Therefore, how to control the catalyst reconstruction process to avoid phase separation caused by deep reconstructing becomes a key challenge to obtaining a highly active OER catalyst.

The other major challenge in seawater splitting is the chlorine evolution reaction (CER) on the anode due to the existence of chloride ions (Cl⁻) in seawater, which competes with OER[7]. Chlorine generated

[1]Key Laboratory of Carbon Materials of Zhejiang Province, College of Chemistry and Materials Engineering, Wenzhou University, Wenzhou 325035, China. ✉e-mail: guody@wzu.edu.cn; xianchen@wzu.edu.cn; shunwang@wzu.edu.cn

from the CER would further react with OH⁻ to yield hypochlorite[2]. In addition, some insoluble precipitates, such as magnesium hydroxide tend to be formed on the electrode surface[28]. These not only raise the overpotential and reduce the efficiency of seawater electrolysis but also significantly deteriorate the structure and properties of the catalyst[8,17,29]. Moreover, it is still difficult to monitor the structural evolution of the catalyst and clarify the properties of the catalytically active surface in the process of electrolysis of seawater[30,31]. Therefore, it is highly desirable to develop OER catalysts with selective inhibition of CER and shielding from impurities in seawater for boosting the exploration of large-scale seawater electrolysis.

To solve the above critical issues, the ultra-thin amorphous molybdenum oxide ($MoO_3$) layer was introduced into the ordered beaded-like cobalt oxide (CoO) array on the three-dimensional carbon cloth (CC) via atomic layer deposition (ALD) technology, thus forming a cowpea-shaped structure catalyst (denoted as $MoO_3$@CoO/CC). The overpotential and interfacial activity can be greatly optimized by accurately regulating the surface of CoO by ALD $MoO_3$, thus accurately affecting the formation process of O* and OOH*, optimizing the reaction mechanism, and improving the kinetics of OER. More importantly, $MoO_3$, as a directional confine layer, inhibits the phase segregation of Co-Mo bimetallic layered double hydroxide (CoMo-LDH) formed by dynamic self-reconfiguration of the catalyst interface, thereby improving the service life of the catalyst. Interestingly, the $MoO_3$ layer can shield Cl⁻ from reaching the catalytic active interface, and the

reconstructed stable CoMo-LDH layer relies on electrostatic repulsion to further hydrophobic chlorine, thus achieving selective seawater oxidation. The flow electrolytic cell composed of $MoO_3$@CoO/CC and commercial Pt/C shows superior hydrogen production performance. At 25 °C, the industrial current (1 A cm⁻²) only requires a voltage of 1.93 V, the hydrogen evolution lasts for 500 h without attenuation, and the Faraday efficiency (FE) remains above 95%, showing high stability and selectivity. In addition, the hydrogen production rate is 419.4 mL cm⁻² h⁻¹, and the power consumption is only 4.62 kWh m⁻³ H₂, showing superior application prospects. This strategy of constructing a directional confine reconstruction layer and selecting a catalytic layer by ALD technology provides a direction for developing cheap and high-performance direct electrolysis seawater catalysts.

## Results

### Synthesis and characterization

Co(OH)F was firstly prepared on a CC substrate by a hydrothermal method as a precursor to obtain beaded-like CoO via annealing treatment. Subsequently, an ultra-thin $MoO_3$ layer was fabricated on the bead-like CoO surface by ALD technology, named $MoO_3$@CoO/CC (Fig. 1a). Inspired by the unique structure of cowpea (Fig. 1b), this work was designed to construct a micron-sized active center and selective confinement layers for a high-performance electrolytic seawater oxidation catalyst. As shown in Fig. 1c, the creation of $MoO_3$@CoO/CC with cowpea-like heterostructures will establish the catalytic

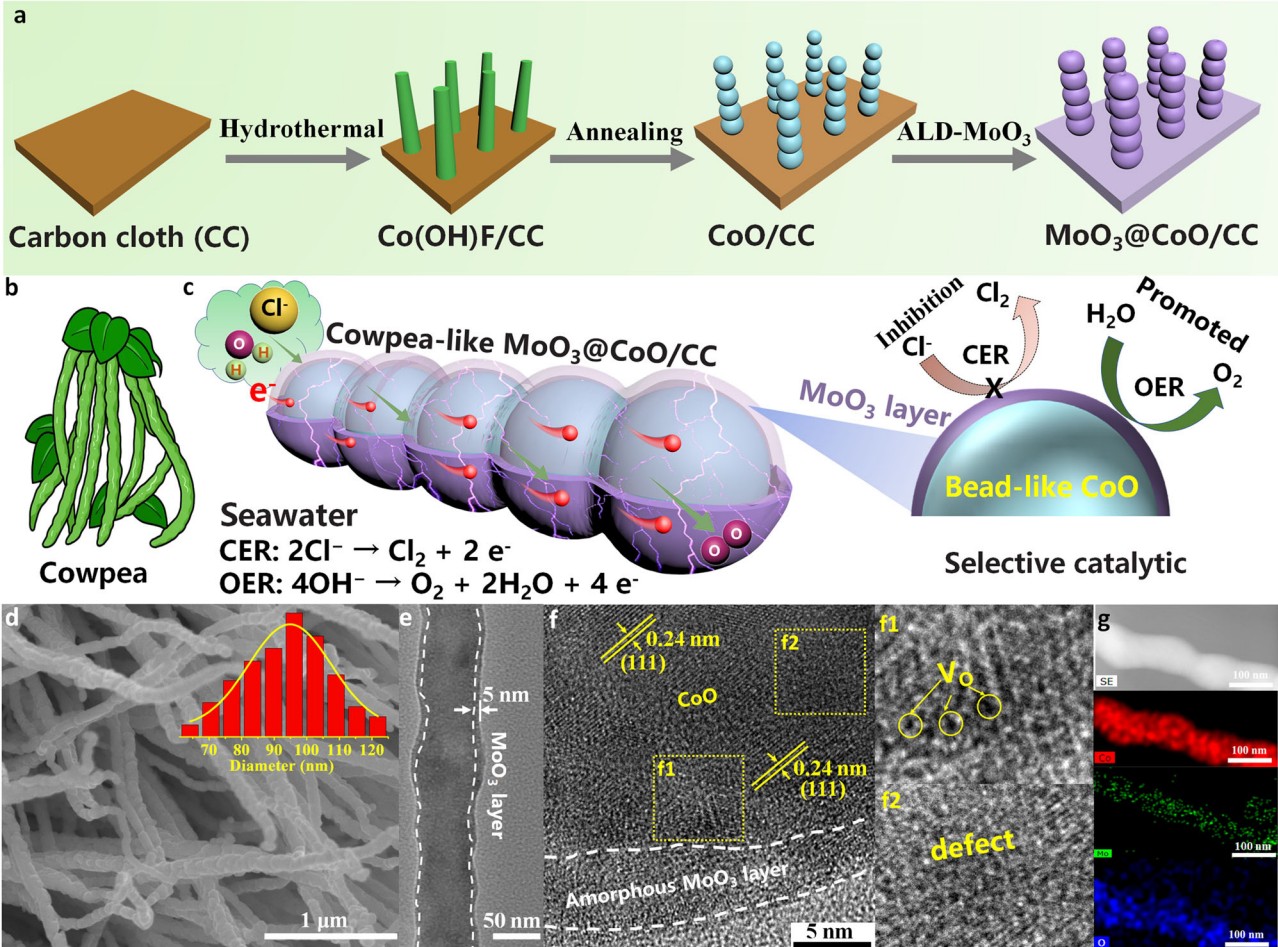

**Fig. 1 | Morphology characterization of prepared samples. a** Schematic illustration of the synthesis process of $MoO_3$@CoO/CC. **b** Cowpea pictures. **c** Schematic illustration of $MoO_3$@CoO/CC with high efficiency and selective oxidation of seawater. **d** SEM, (**e**) TEM, (**f**) HRTEM, (**g**) EDS-mapping of $MoO_3$@CoO/CC. The illustration in **d** is the diameter statistics of $MoO_3$@CoO/CC.

conversion process of selective adsorption of $H_2O$ in seawater electrolysis, thus targeting and promoting the OER process in seawater.

The beaded-like CoO with abundant defects (Supplementary Fig. 1 and Supplementary Fig. 2 of the Supplementary Information for the detailed analysis process) and exposed oxygen in the structure provide rich active sites for ALD[32,33]. The $MoO_3$ prepared by using $Mo(CO)_6$ and oxygen plasma as an ALD reaction source is anchored on the surface of CoO by Co–O–Mo bonds. Compared with CoO, the morphology of $MoO_3@CoO/CC$ has not been changed obviously (Fig. 1d and Supplementary Fig. 3), which indicates that the $MoO_3$ layer deposited by ALD exhibits good shape retention[34,35]. However, the diameter of cowpea-like $MoO_3@CoO/CC$ increases to ~95 nm (inset Fig. 1d). A $MoO_3$ layer of about 5 nm is observed on the beaded-like CoO surface (Fig. 1e). The results are consistent with statistical analysis (The inset in Fig. 1d and Supplementary Fig. 2b). Figure 1f shows that the lattice spacing is 0.24 nm, which corresponds to the (111) plane of CoO nanocrystal. Notably, a large number of oxygen vacancies ($V_O$) and defects are observed in the f1 and f2 regions of the CoO, implying its good catalytic activity. The EDS-Mapping spectrum displays the uniform distribution of Mo, O, and Co on the entire skeleton (Fig. 1g).

## Structural analysis

In Fig. 2a, the X-ray diffraction (XRD) pattern of beaded-like CoO/CC is very consistent with the standard peak of the CoO cubic crystal phase (JCPDS No. 48–1719)[36]. The characteristic diffraction pattern of $MoO_3$ was not detected by XRD, demonstrating that the $MoO_3$ layer is amorphous. The diffraction pattern of CoO in $MoO_3@CoO/CC$ samples shifted negatively, indicating that there is a strong chemical force between $MoO_3$ and CoO. From Fig. 2b, the Raman characteristic peaks at 482, 523, and 684 $cm^{-1}$ are attributed to CoO[15]. Compared with CoO, the characteristic peak of CoO in $MoO_3@CoO/CC$ shows an obvious blue shift, which further indicates that there is a strong electronic interaction between $MoO_3$ and CoO.

The chemical states of various elements in $MoO_3$ and CoO (Supplementary Fig. 4) were analyzed by X-ray photoelectron spectroscopy (XPS). In Fig. 2c, the high-resolution XPS spectrum of Co 2p shows that the binding energies of 780.8 and 796.5 eV correspond to $Co^{2+}$ $2p_{3/2}$ and $Co^{2+}$ $2p_{1/2}$[37,38], respectively. After $MoO_3$ is deposited on the surface of CoO, the characteristic peaks of Co $2p_{3/2}$ and Co $2p_{1/2}$ shift negatively, indicating that a chemical bond was formed between CoO and $MoO_3$. Compared with $MoO_3/CC$, the binding energies of Mo $3d_{5/2}$ and Mo $3d_{3/2}$ in $MoO_3@CoO/CC$ shift negatively (Fig. 2d), implying that the substrate CoO affects the chemical state of Mo. In Fig. 2e, there are obvious peaks at 529.9 eV, which are attributed to the binding energy of metal–oxygen bond (M–O), and the M–O bond in $MoO_3@CoO/CC$ exhibits a significant blue shift[39,40]. The peak of $MoO_3@CoO/CC$ is stronger at 531.4 and 532.3 eV (–OH and $O_{ads}$). These results indicate that $MoO_3$ has been successfully anchored on the CoO interface by the Co–O–Mo bond, and the adsorption capacity of the CoO interface has been improved.

From the electron paramagnetic resonance (EPR) spectrum, CoO/CC and $MoO_3@CoO/CC$ samples show obvious signals at the position of g = 2.002 (Fig. 2f), indicating that there are rich-$V_O$[39] Compared with CoO/CC, $MoO_3@CoO/CC$ shows significantly increased EPR intensity. This is mainly due that the oxygen plasma can not only be used as a co-reactant for the preparation of $MoO_3$ by ALD but also produce more oxygen defects on the $MoO_3@CoO/CC$.

## Electrochemical test

The electrocatalytic OER of as-prepared was studied in artificial seawater (1 M KOH + 0.5 M NaCl) with a three-electrode system. Powder $MoO_3$ with the same content was constructed on CoO/CC by a physical coating method, designated $MoO_3$-CoO/CC. In Fig. 3a, b, $MoO_3@CoO/CC$ exhibits lower overpotentials than other preparation materials and commercial $RuO_2/CC$ catalysts, suggesting its higher OER electrocatalytic activity. Interestingly, the $MoO_3@CoO/CC$ achieved 800 mA $cm^{-2}$ at an overpotential of only 650 mV. The above results show that the ALD $MoO_3$ layer plays a key role in improving OER performance in artificial seawater. In addition, with the increase of deposition cycles, the catalytic activity and chlorine shielding ability of OER first increased and then decreased (Supplementary Fig. 5). The OER exhibits maximum catalytic activity after deposition of $MoO_3$ for 500 cycles (thickness c.a. 5 nm). Subsequently, the catalytic activity of the catalyst decreases with increasing the number of deposition

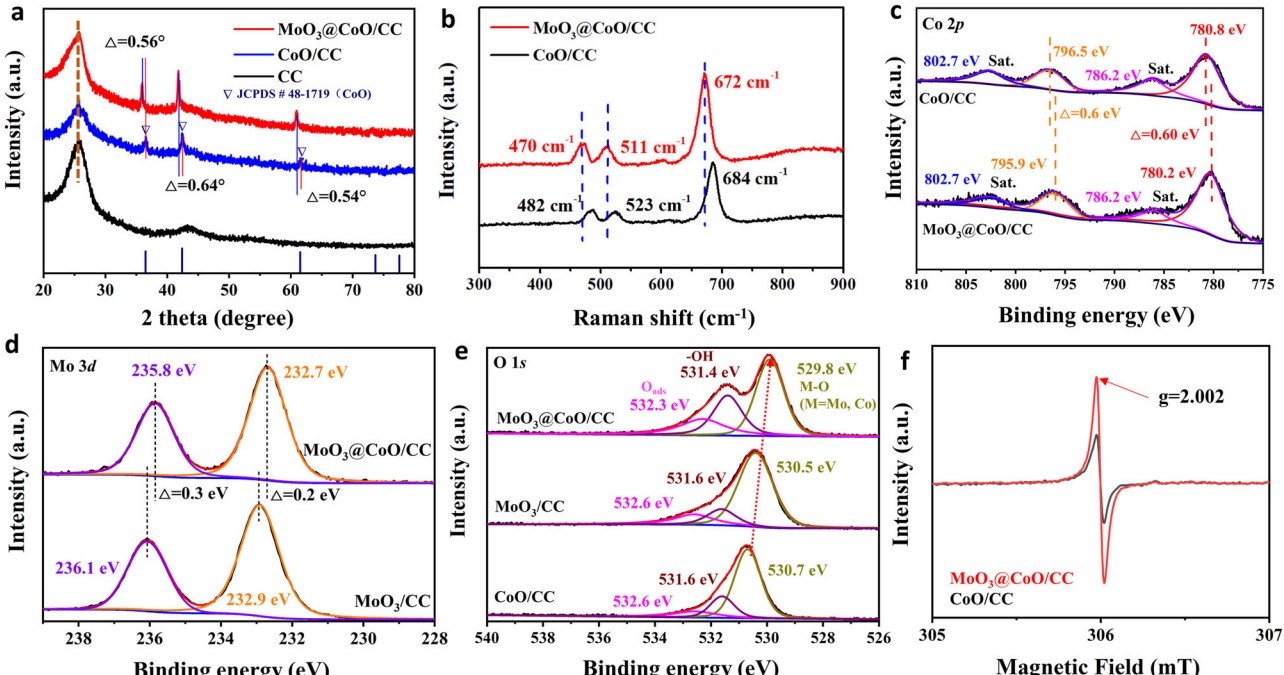

**Fig. 2 | Structural analysis of prepared samples. a** XRD patterns, (**b**) Raman spectra of various samples. The high-resolution XPS spectra of **c** Co 2p, (**d**) Mo 3d, and **e** O 1s for the as-prepared catalysts. The satellite is denoted as "Sat." **f** The EPR spectra of $MoO_3@CoO/CC$ and CoO/CC catalysts.

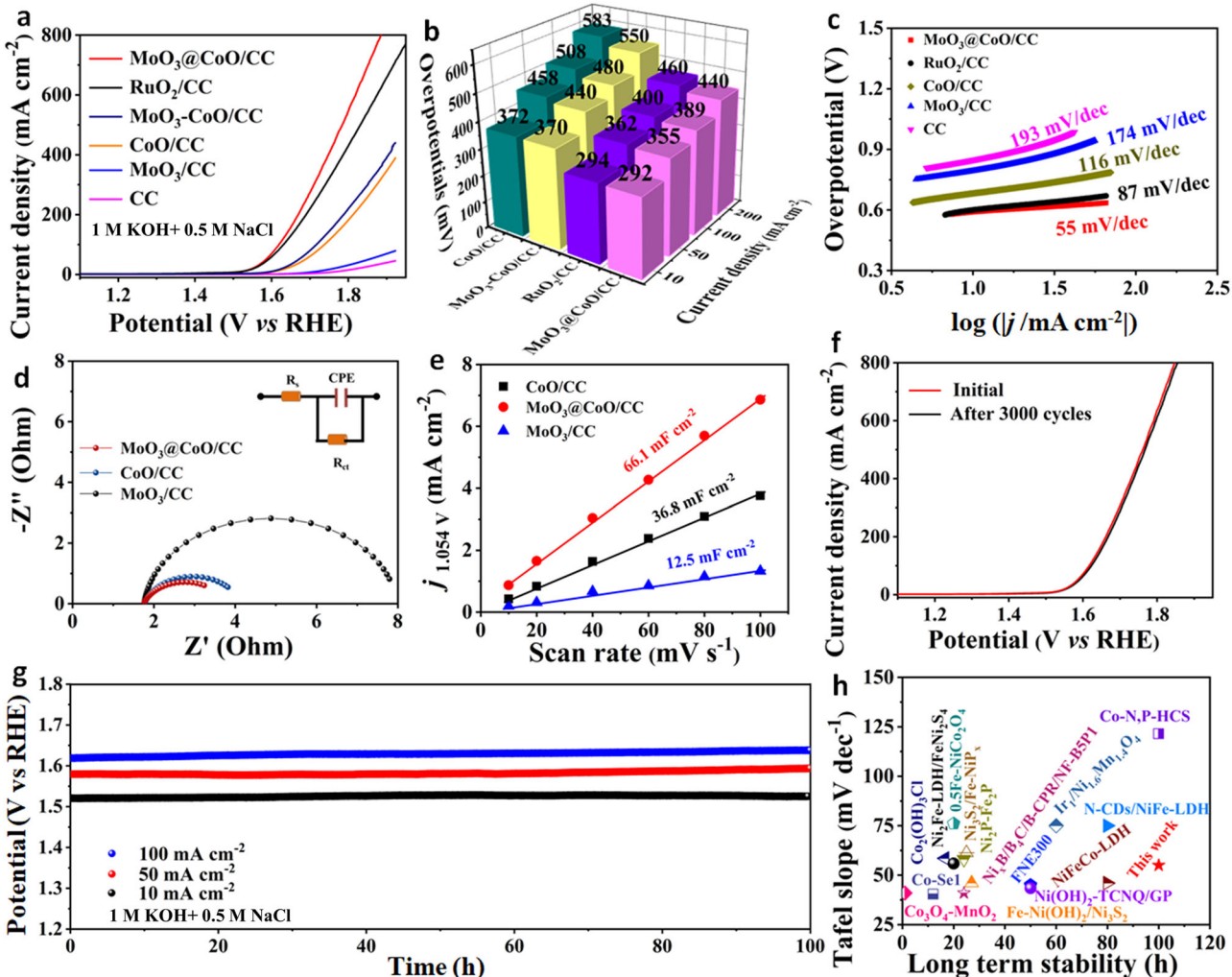

**Fig. 3 | The electrochemical performance of the material in 1 M KOH + 0.5 M NaCl electrolyte was investigated. a** LSV curves of various samples at 20 mV s$^{-1}$; The mass of the catalyst is kept at about 1.89 mg cm$^{-2}$. **b** Overpotential of various samples at different current densities. **c** Tafel slopes. **d** EIS curves. **e** Plots of capacitive currents vs different scan rates with calculated $C_{dl}$; (**f**) LSV cures before and after 3000 CV cycles of MoO$_3$@CoO/CC. **g** Evaluation of MoO$_3$@CoO/CC stability by chronoamperometry method. **h** Tafel slope and long-term stability compared with the literature. The Tafel slopes and long-term stability data are from references (Supplementary Table 2).

cycles. This decrease can be attributed to the excessively thick shielding layer, which reduces the active area of the catalyst (Supplementary Fig. 6). These results imply that the catalyst interface serves as the catalytic active center. From Fig. 3c, the Tafel slope (55 mV dec$^{-1}$) of MoO$_3$@CoO/CC is significantly lower than that of the other samples, indicating its fastest reaction kinetics for the electrolysis of artificial seawater. In addition, a low charge transfer resistance is achieved for MoO$_3$@CoO/CC ($R_{ct}$ = 1.9 Ω), which further highlights its fast reaction kinetics (Fig. 3d and Supplementary Table 1). The electrochemical active surface area (ESCA) of the catalyst was evaluated by measuring the cyclic voltammogram (CV) of the as-prepared catalyst (Supplementary Fig. 7) and calculating its electric double-layer capacitance ($C_{dl}$). From Fig. 3e, the $C_{dl}$ of MoO$_3$@CoO/CC, CoO/CC, and MoO$_3$/CC are calculated to be 58.4, 20.0, and 11.1 mF cm$^{-2}$, respectively. Based on this, it is inferred that MoO$_3$@CoO/CC possesses a relatively large ESCA, implying its rich catalytic active sites. The linear sweep voltammetry (LSV) curve of the catalyst is normalized according to the ESCA (Supplementary Fig. 8). Compared with CoO/CC and MoO$_3$/CC, MoO$_3$@CoO/CC possesses the highest current density and the lowest overpotential, demonstrating its highest intrinsic catalytic activity. The highest turnover frequency value of the MoO$_3$@CoO/CC catalyst obtained further supports the above result (Supplementary Fig. 9).

The stability of MoO$_3$@CoO/CC under various potentials was investigated. The LSV curve remained almost unchanged after 3000 CV cycles of continuous reaction, showing its superior stability (Fig. 3f). Furthermore, the continuous OER stability of MoO$_3$@CoO/CC catalyst at varying current densities using chronoamperometry (Fig. 3g). These overpotentials are almost unchanged after 100 h, which further indicates that the MoO$_3$@CoO/CC catalyst exhibits good durable stability. More importantly, the MoO$_3$@CoO/CC catalyst possesses high reaction kinetics and lasting stability compared with the other reported non-noble metal catalysts (Fig. 3h and Supplementary Table 2).

### Mechanism analysis of inhibiting CER

This superior catalytic activity and stability of MoO$_3$@CoO/CC catalyst may be attributed to the tailored MoO$_3$ layer, which possesses the effects of shielding Cl$^-$ and regulating interfaces (Fig. 4a). To verify this hypothesis, formation energy, migration path, and migration energy barrier between Cl$^-$ and catalyst in seawater were calculated by density functional theory (DFT) (Supplementary Fig. 10). Compared with the formation energy of CoO/CC and Cl$^-$ (−3.3 eV), the formation energy of MoO$_3$@CoO/CC and Cl$^-$ is only −0.41 eV (Fig. 4b). Compared with Cl$^-$, OH$^-$ with smaller radius easily passes through MoO$_3$ layer[41]. In Fig. 4c, a

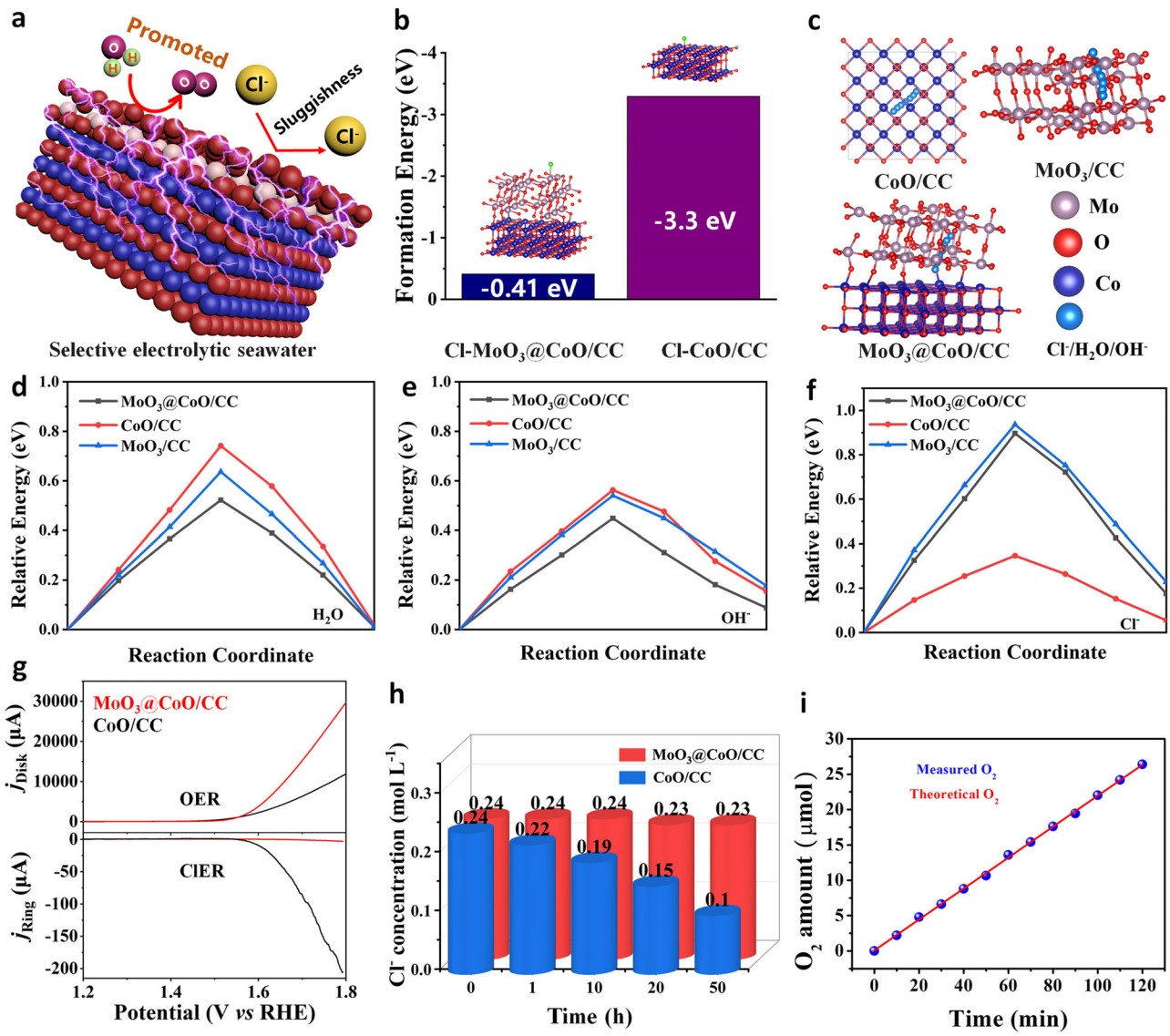

**Fig. 4 | Mechanism analysis of inhibiting CER. a** Schematic illustration of Cl⁻ barrier mechanism. **b** The formation energy of CoO/CC and MoO₃@CoO/CC with chlorine is calculated according to DFT calculate. **c** the migration paths of Cl⁻, H₂O, and OH⁻ in various structure. The migration energy barrier of **d** H₂O, (**e**) OH, and **f** Cl in various structures. **g** Evaluation of OER and CER of catalysts by RRDE. **h** The Cl⁻ concentration in that electrolyte was measured by the Cl⁻ detector after 50 h of continuous OER reaction for various catalysts. **i** The amount of O₂ collected in a Pt/C/CC||MoO₃@CoO/CC two-electrode cell compared to the theoretical gas product was in 1 M KOH + 0.5 M NaCl electrolyte at 20 mA cm⁻².

structural model of the migration path of Cl⁻/H₂O/OH⁻ in various samples is constructed. As assumed, the migration energy barrier of H₂O/OH at MoO₃@CoO/CC interface is lower than that at CoO/CC and MoO₃/CC interface (Fig. 4d, e). Compared with CoO interface, the migration energy barrier of Cl⁻ at MoO₃@CoO/CC and MoO₃/CC interface is higher (Fig. 4f). Calculation of the change in differential charge at the interface of various samples after adsorption of H₂O/OH⁻/Cl⁻, respectively, further demonstrates that MoO₃@CoO/CC is highly active towards H₂O/OH⁻ and shows inertness toward Cl⁻ (Supplementary Fig. 11). These results verify that MoO₃ layer exhibits the ability to shield Cl⁻, which provides the possibility for selective catalysis.

The competition between OER and CER of the target catalyst was further evaluated via the voltammetry experiment of a rotating ring-disk electrode (RRDE)[42]. Compared with CoO/CC, MoO₃@CoO/CC exhibits almost no CER in artificial seawater (Fig. 4g), which indicates that the ultra-thin MoO₃ layer effectively obstructs Cl⁻ to achieve selective catalysis. Moreover, the change in Cl⁻ concentration in the electrolyte after 50 h of OER on various catalysts was tested by Cl⁻

detector (Fig. 4h). The Cl⁻ concentration in the electrolyte of MoO₃@CoO/CC remained almost unchanged compared to CoO/CC. In alkaline artificial seawater, the O₂ production FE of MoO₃@CoO/CC catalyst is close to 100% (Fig. 4i), which further indicates its high selectivity in the electrolysis of seawater.

## Restricted dynamic surface self-reconstruction

The MoO₃@CoO/CC catalyst still maintained a cowpea-like structure, but its surface became rough ((Fig. 5a and Supplementary Fig. 12). The thickness (~5 nm) of the MoO₃ layer has not changed significantly, indicating that the MoO₃ layer has not been reconstructed (Fig. 5b). Obviously, darker areas were observed between MoO₃ layer and CoO (Compared with Fig. 1e), and the corresponding spacing increased significantly, indicating that a new phase was formed between them. Importantly, a lattice spacing of 0.38 nm was observed between the MoO₃ layer and CoO (Fig. 5b1), corresponding to the (006) crystal plane of CoMo-LDH[43]. From Fig. 5b2, the lattice of CoO underwent distortion due to the compression resulting from the reconstruction of the interface[44]. Fortunately, CoO is still rich in

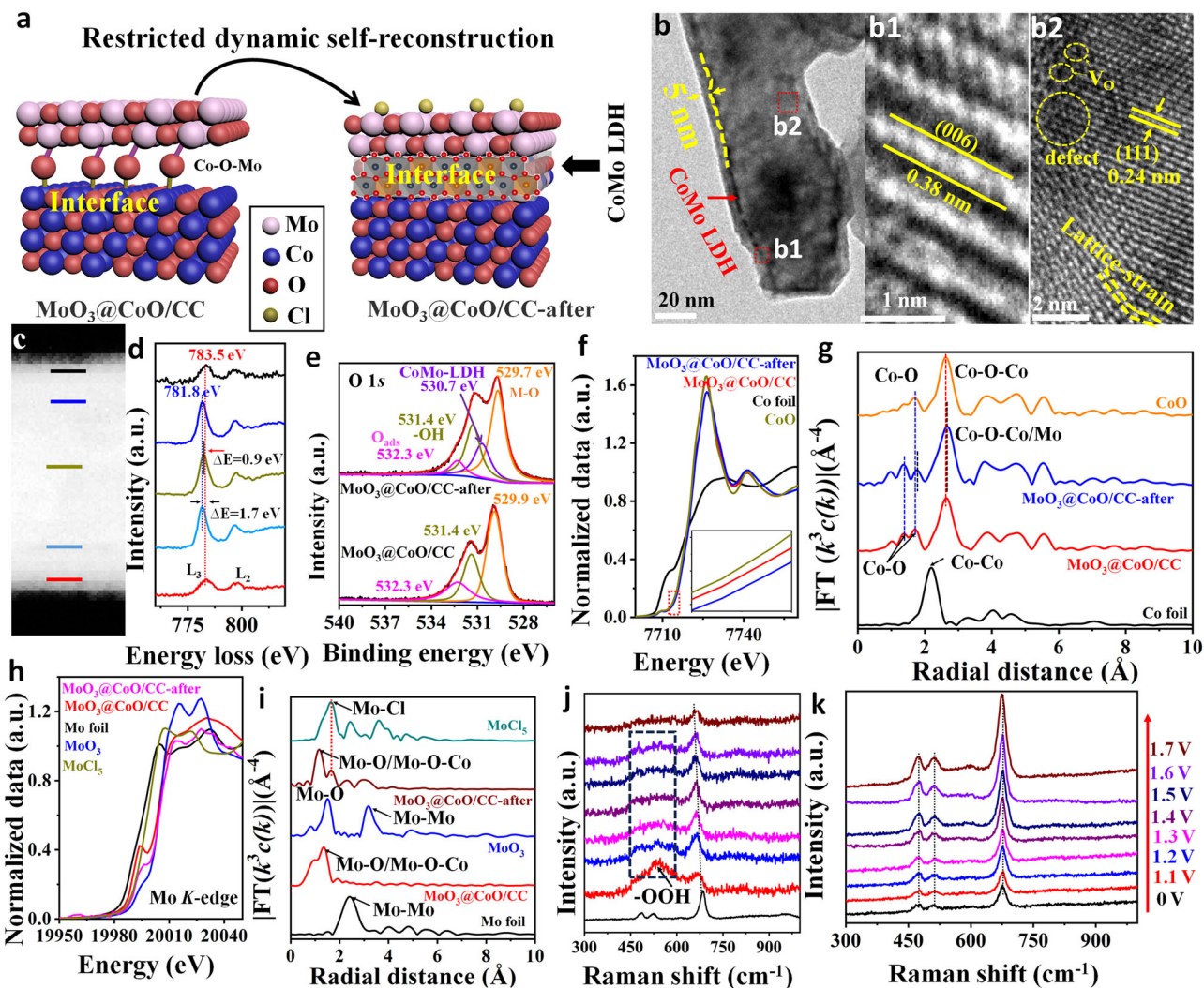

**Fig. 5 | Mechanism analysis of interface restricted dynamic self-reconstruction. a** Schematic diagram of interface reconfiguration. **b** TEM of MoO₃@CoO/CC catalyst after 50 h of continuous OER in artificial seawater (designated MoO₃@CoO/CC-after), b1 and b2 is the HRTEM in the red box in **b**. **c** EELS Spectrum image (low-loss). **d** EELS spectra of Co-L-edge. **e** High-resolution XPS spectra of O 1s. **f** XANES curves of the Co K-edge. **g** FT-EXAFS curves of various Co-based samples. **h** XANES curves of the Mo K-edge. **i** FT-EXAFS curves of various Mo-based samples. In situ, the Raman spectrum of the **j** CoO/CC and **k** MoO₃@CoO/CC in 1 M KOH + 0.5 M NaCl was measured at different potentials.

$V_O$ and defects, indicating that the interface self-reaction is accurately controlled without causing deep reconstruction. Verification of reconfiguration to form CoMo-LDH by testing the electron energy loss spectrum (EELS) of the interfacial layer of MoO₃@CoO-after (Fig. 5c). From Fig. 5d, the Co-L3 edge of the interfacial layer is blue-shifted by about 0.6 eV compared to the intermediate-phase CoO, indicating a higher Co valence[24]. Moreover, the EELS of O and Mo further verified the formation of the CoMo-LDH phase between MoO₃ and CoO (see Supplementary Fig. 13 for detailed analysis). Interestingly, the active area of the MoO₃@CoO/CC sample did not decrease after OER. Conversely, the active area of the CoO/CC catalyst decreases significantly (Supplementary Fig. 14). In addition, the XPS of the corresponding Co 2p and Mo 3d displays an obvious blue shift (Supplementary Fig. 15), indicating that a new phase is formed after OER. The high-resolution XPS analysis of O 1s shows that the Co-O bond is strengthened and a new bond (530.7 eV) appears[10,45], which suggests that the CoMo-LDH phase is reconstructed during OER (Fig.5e). The CoMo-LDH layered compound formed by this reconstruction is composed of MO₆ (M=Co, Mo) octahedral structure, which has strong electrostatic repulsion to Cl⁻, thus further realizing selective catalysis.

Figure 5f shows the Co-k edge X-ray absorption near edge structures (XANES) for various samples. The absorption edge of MoO₃@CoO/CC is located near CoO (inset in Fig. 5f), indicating that the Co valence state in MoO₃@CoO/CC is close to that of CoO. This is mainly due to the formation of chemical bonds between MoO₃ and CoO, which makes the Co valence state in MoO₃@CoO/CC higher than that of pure CoO. This further verified that MoO₃ was anchored on the CoO surface by the Co–O–Mo bond. In addition, the Co valence state in MoO₃@CoO/CC-after is obviously higher than that in MoO₃@CoO/CC, which is attributed to the formation of CoMo-LDH after OER. The coordination peaks of Co–O and Co–O–Co in the CoO reference samples were found to be 1.38 and 2.67 Å in the Fourier transform (FT) k³-weighted extended X-ray absorptiometry (EXAFS) spectrum (Fig. 5g). The K-space (Supplementary Fig. 16) and lattice structure (Supplementary Fig. 17) verify the above fitting results. Compared with CoO, the Co–O and Co-O-Co bonds of MoO₃@CoO/CC and MoO₃@CoO/CC-after samples red-shifted, respectively. Note that since the key signals of Co-O-Co and Co-O-Mo are close, they are collectively referred to as Co-O-Co/Mo. Meanwhile, the Co–O bond of MoO₃@CoO/CC splits into two peaks, which is related to the formation of the Co–O–Mo bond[13]. Moreover, the splitting peak signal at 1.38 Å is

stronger than the main peak (1.71 Å), indicating the formation of the Co-O-O bond. The Co-O-Mo bond of MoO₃@CoO/CC-after did not shift compared to MoO₃@CoO/CC, indicating that MoO₃ was firmly anchored to the CoO interface by the Co-O-Mo bond during the OER process. Co K-edge wavelet transform (WT)-EXAFS (Supplementary Fig. 18) and quantitative fitting EXAFS (Supplementary Fig. 19) analysis further verify the above analysis.

The Mo-K edge of MoO₃@CoO/CC-after lies between MoCl₅ and MoO₃ (Fig. 5h), indicating that the valence of Mo has been improved after OER. This may be due to the formation of Mo-Cl coordination bonds. The Mo-O bond in MoO₃@CoO/CC shifts negatively by 0.15 Å compared to MoO₃ (Fig. 5i). This is mainly attributed to the existence of partial Co-O-Mo bonds in MoO₃@CoO/CC. The Mo-O/Co-O-Mo bond in MoO₃@CoO/CC-after shifts negatively by 0.18 Å compared with that of MoO₃@CoO/CC, indicating the formation of the CoMo-LDH phase at the interface. The Mo-Cl bond was detected in MoO₃@CoO/CC-after, indicating that the Cl⁻ was blocked via the outer MoO₃ during the electrolysis of seawater. The specific coordination of Mo atoms was further obtained by quantitatively fitting EXAFS of various samples (Supplementary Fig. 20 and Supplementary Table 3). WT-EXAFS analysis showed that the strongest signal region of MoO₃@CoO/CC-after became irregular compared with MoO₃@CoO/CC, which further indicated that new Mo-O-O and Mo-Cl bonds were formed on the catalyst after OER (Supplementary Fig. 21). Thus, the above analysis demonstrates that the directional reconfiguration of the CoMo-LDH phase in the presence of a confined MoO₃ layer, synergistically improves the Cl⁻ shielding ability and increases the catalyst activity and stability.

In situ, Raman spectroscopy (Fig. 5j) showed that the application of voltage to the CoO/CC catalyst resulted in the disappearance of its characteristic CoO peaks (at 470, 511, and 672 cm⁻¹), whereas the CoOOH peaks became evident[39]. In addition, the characteristic peak of CoOOH gradually weakens with the increase in voltage, which is mainly due to the generation of the phase-separated bulk catalyst with the occurrence of deep reconstruction. In contrast, in the OER reaction of MoO₃@CoO/CC under various voltages, the characteristic peak of the catalyst remains unchanged, indicating that the deep restructuring of the catalyst is inhibited by the MoO₃ confinement effect. Interestingly, the characteristic peak of MoO₃@CoO/CC is obviously blue-shifted with the increase of voltage (Fig. 5k), indicating that the catalyst interface is reconstructed directionally. Therefore, the reconstruction of the MoO₃@CoO/CC catalyst can be persistently maintained through the confinement of the MoO₃ layer, thus inhibiting the unlimited-depth reconstruction (Supplementary Fig. 22), which results in superior stability of this catalyst and achieves accurate control of the active interface reconstruction.

## Mechanism analysis of seawater oxidation

Figure 6a shows that the ultra-thin MoO₃ barrier layer does not impact the mass transfer of the catalyst during the catalytic process (Supplementary Fig. 23). From Fig. 6b, the O−H bond of water adsorbed by CoO/CC catalyst corresponds to the wavelength of 1638 cm⁻¹ [13]. Compared with CoO/CC, the O−H bond on MoO₃@CoO/CC is obviously blue-shifted, indicating that MoO₃@CoO/CC effectively promotes H₂O adsorption and activation. The corresponding model is constructed by DFT to further verify that the ultra-thin MoO₃ layer regulates the adsorption and activation of H₂O/OH⁻ on the CoO interface (Fig. 6c, Supplementary Fig. 24 and Supplementary Fig. 25). Besides, the adsorption energy of *Cl on MoO₃@CoO/CC (−0.19 eV) is significantly lower than that of CoO (−0.67 eV). In Fig. 6d, MoO₃@CoO/CC shows a higher local state than CoO/CC, indicating that MoO₃@CoO/CC possesses more electron concentration on the heterointerfaces during the process of OER[46]. In addition, the differential charge accumulation and dissipation in MoO₃@CoO/CC are more pronounced than those in CoO/CC,

indicating that there is significant charge transfer favoring the catalytic reaction (Supplementary Fig. 26).

Interestingly, compared with MoO₃@CoO/CC, MoO₃@CoO/CC-after shows a significant increase in EPR intensity, indicating an increase in V_O during OER (Fig. 6e). This may be due to the selective-confinement modulation of MoO₃ layer (Supplementary Fig. 27), which transforms the adsorbate evolution mechanism (AEM) into lattice-oxygen-mediated mechanism (LOM)[47,48]. The catalytic mechanism of the catalyst was further verified by pDOS of oxygen, tetra-methylammonium cation (TMA⁺) detection, differential electro-chemical mass spectrometry (DEMS), and pH-dependent experiments for the catalytic mechanism changed from AEM to LOM after directed reconfiguration. From Fig. 6f, the pDOS further implies that O−O coupling effectively eliminates unpaired O atoms of oxygen around the Fermi level[48]. The above results are consistent with Co k-edge FT-EXAFS analysis. It is very important to detect the O₂²⁻ species produced by LOM during OER by TMA⁺ [49,50]. The OER activity of the MoO₃@CoO/CC catalyst was significantly reduced after the addition of TMAOH to the alkaline electrolyte (Supplementary Fig. 28). This is mainly due to the strong interaction between O₂²⁻ species and TMA⁺, which inhibits the LOM pathway of the catalyst. On the contrary, because the CoO/CC catalyst evolves oxygen by the AEM mechanism, thus the performance changes are not obvious[51]. From Fig. 6g, it is observed that MoO₃@CoO/CC catalyst exhibits two characteristic peaks corresponding to TMA⁺ at 751.7 and 950.6 cm⁻¹, which further proves that its OER process follows the LOM mechanism[50]. In addition, the OER activity of MoO₃@CoO/CC catalyst at different pH values was significantly enhanced with the increase of pH value compared to CoO/CC (Fig. 6h). The RHE-scaled proton reaction level ($\rho^{RHE} = \partial \log(j)/\partial pH$) was used to further elucidate the dependence of the catalyst undergoing an OER reaction on proton activity[50]. The $\rho^{RHE}$ value of MoO₃@CoO/CC (0.71) is closer to 1 than that of CoO/CC (0.12), which further indicates the pH dependence of OER kinetics (Fig. 6i), thus proving that the reconstructed MoO₃@CoO/CC follows LOM rather than the traditional AEM in the OER process[50]. To clarify the OER process of LOM directly, the activated MoO₃@CoO/CC catalyst was labeled with an ¹⁸O isotope. The results show an obvious periodic intensity of the ¹⁸O¹⁶O peak, while ¹⁸O¹⁸O exhibits no signal (Fig. 6j), which further verified that the MoO₃@CoO/CC catalyst mechanism that undergoes activation-directed reconfiguration transforms into LOM[52,53]. Moreover, this reaction mechanism was further verified by calculating Gibbs free energy (ΔG) of MoO₃@CoO/CC and CoO/CC (Fig. 6k). The ΔG to form *OH and *OOH in the catalyst system is reduced by the anchoring of MoO₃ at the CoO interface to trigger AEM (Supplementary Table 4). Initiating surface reconstruction in a catalyst with an external potential results in a significant reduction in the ΔG to form *O in favor of O₂, thereby triggering the LOM.

## Practical electrolysis applications

The concentration of salt will gradually increase with the continuous electrolysis of seawater in practical electrolysis applications[54]. Fig. 7a shows that the MoO₃@CoO/CC achieved a current density of 400 mA cm⁻² with overpotentials of only 505, 498, 531, and 552 mV in 1 M NaCl, 1.5 M NaCl, alkaline real seawater and real seawater, respectively. Even at 800 mA cm⁻², the MoO₃@CoO/CC catalyst still exhibits a low overpotential in various electrolytes (Supplementary Table 5). This further proves that this catalyst possesses high selectivity for electrolysis of seawater in high-concentration seawater. The mass spectrum shows that there is no Cl₂ precipitation, and the gas chromatography shows that the relative FE of O₂ generation is 100% (Supplementary Fig. 29), which further verifies that the MoO₃ layer can effectively shield Cl⁻, thus realizing high-efficiency selective seawater OER. As shown in Fig. 7b, no significant increase in voltage was observed by chronopotentiometry evaluation of the catalyst for continuous OER in high-salinity artificial seawater and real seawater for 1000 h, indicating

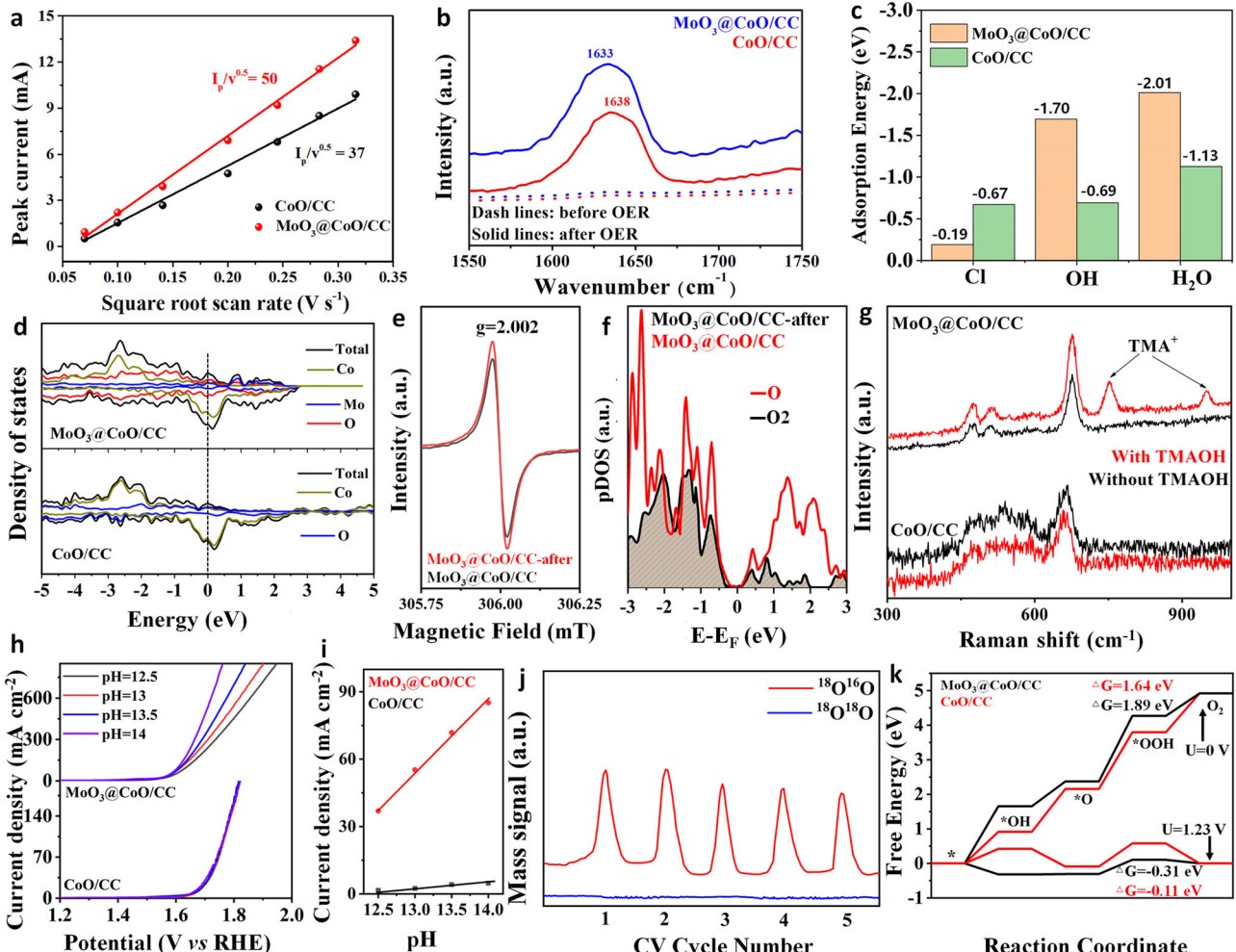

**Fig. 6 | Mechanism analysis of promoting OER reaction. a** Plots of $I_P$ derived from the peak as a function of $v^{1/2}$ for different electrodes. **b** FT-IR ATR spectra of various catalysts after electrolysis for 20 h at 20 mA cm$^{-2}$. **c** The adsorption energy of various samples. **d** The DOS curves of various catalysts. **e** The EPR spectra of various catalysts. **f** pDOS of O(2p) orbitals in the L2 intermediate for various catalysts, respectively. **g** Raman spectrum of various catalysts tested after washing with water after testing with or without TMAOH. **h** LSV curves of various samples at different pH values. **i** The proton reaction order was calculated according to $\rho^{RHE} = \partial log(j)/\partial pH$ at 1.6 V vs. RHE. **j** DEMS isotope labeling test of MoO$_3$@CoO/CC. **k** Calculated free energy of OER intermediates at 0 and 1.23 V.

that the MoO$_3$@CoO/CC possesses lasting stability. In addition, the current density for this catalyst can reach 600 mAcm$^{-2}$ in 6 M KOH + 1.5 M NaCl at 60 °C with only 1.70 V, and hardly decays duration of 1000 h. This performance is significantly higher than that reported in the literature (Supplementary Table 6). This well meets the operational demands of industry electrolytic cells in high temperatures and concentrated alkali conditions.

We assembled a flowing real seawater electrolyzer to evaluate the feasibility of large-scale H$_2$ production using MoO$_3$@CoO/CC, Pt/C, and amphoteric ionic membranes as anode, cathode, and separation, respectively (Fig. 7c). Notably, seawater enters the flow cell from the anode MoO$_3$@CoO/CC electrode side. Figure. 7d shows the polarization curves of MoO$_3$@CoO/CC||Pt/C/CC cell at different temperatures and electrolytes. The current density of direct real seawater electrolysis up to 2.0 A cm$^{-2}$ at 60 °C requires only 1.99 V. The results are comparable to state-of-the-art pure water electrolysers[55]. At 25 °C, the cell voltages of high salinity artificial seawater and real seawater are 1.81 and 1.93 V, respectively, which provide the industrial hydrogen production current density of 1.0 A cm$^{-2}$. The catalysts were stable at 1 A cm$^{-2}$ for 500 h with almost no degradation (Fig. 7e) and produced H$_2$ and O$_2$ with an FE of 95% (Fig. 7f), indicating that the as-prepared catalysts have good direct seawater electrolysis stability and high selectivity. In contrast, the electrolytic cell consisting of the RuO$_2$/CC||

Pt/C/CC exhibits poor stability and low selectivity (Supplementary Fig. 30). The above results show that the ALD MoO$_3$ layer effectively blocks the CER in seawater and inhibits the cation from entering the cathode, thus improving the anode selective oxidation and cathode stability (Supplementary Fig. 31). More importantly, the H$_2$ production rate in the MoO$_3$@CoO/CC||Pt/C/CC direct seawater electrolyzer is about 419.4 mL cm$^{-2}$ h$^{-1}$ at 25 °C. The power consumption is only 4.62 KWh m$^{-3}$ H$_2$, which is lower than the energy consumption of pure water electrolysis (~5 kWh m$^{-3}$ H$_2$)[56,57]. The performance of the results far exceeds those reported in real seawater electrolysis (Fig. 7g). Therefore, this research provides a promising solution for the economical and efficient production of H$_2$ by direct electrolysis of seawater.

## Discussion

In summary, we successfully prepared cowpea-like MoO$_3$@CoO/CC catalyst by using ALD technology to construct ultra-thin amorphous MoO$_3$ on the surface of beaded CoO with abundant defects and V$_o$. The MoO$_3$@CoO/CC exhibits a low overpotential of 440 mV at 200 mA cm$^{-2}$, a small Tafel slope of 55 mV dec$^{-1}$, and superior stability for electrolytic seawater OER. This is mainly attributed to the following three advantages: (1) The MoO$_3$ layer and CoMo-LDH phase in-situ formed blocks Cl$^-$ and allows H$_2$O/OH$^-$ to enter the

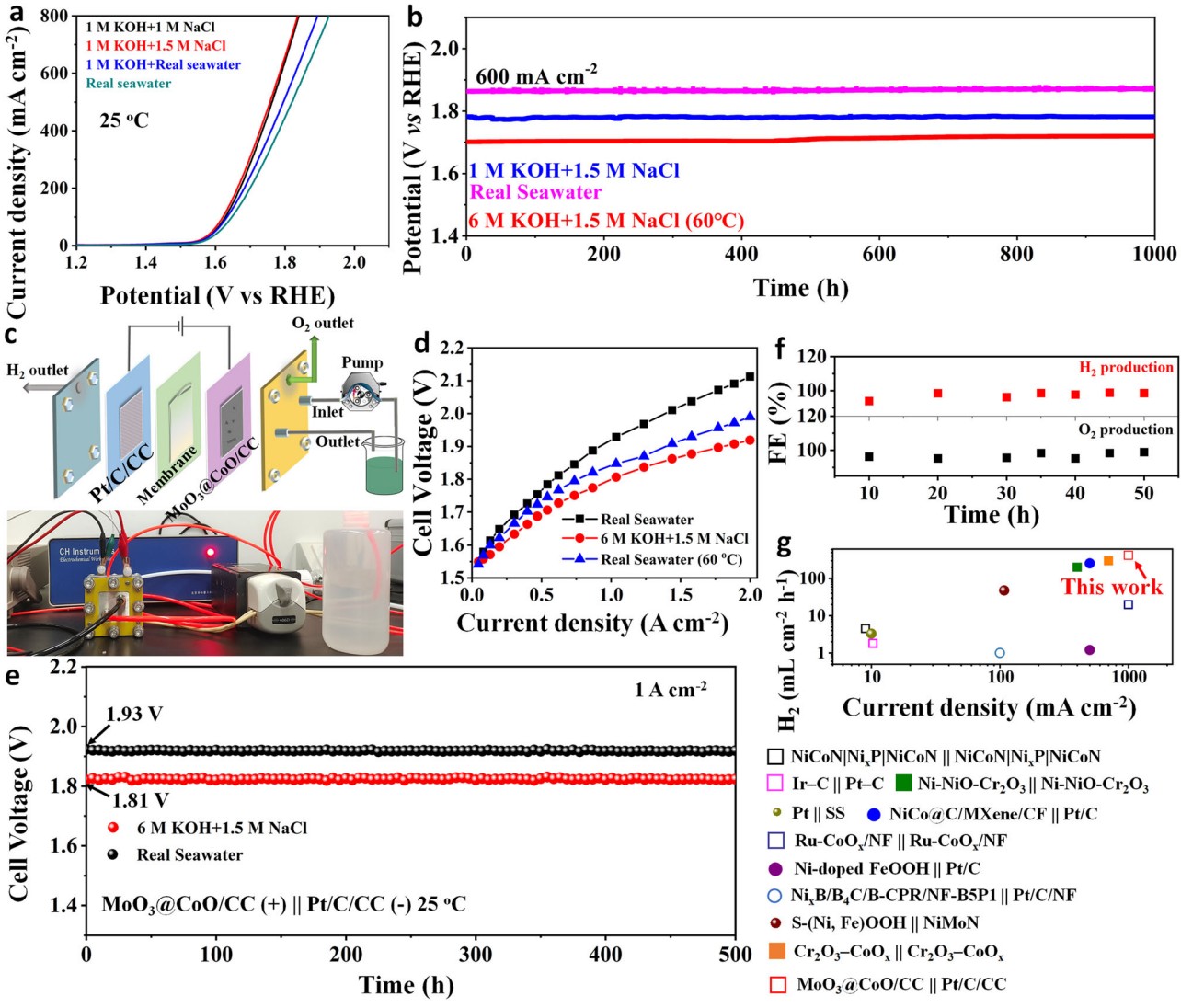

**Fig. 7 | Application of catalyst.** Catalytic performance in different electrolytes: **a** LSV curves of MoO$_3$@CoO/CC catalyst in various electrolytes at 25 °C. **b** The stability of MoO$_3$@CoO/CC catalyst in various media was evaluated by chronopotentiometry. Performance of two-electrode system: (**c**) schematic diagram and optical photograph of flow electrolytic cell. **d** Polarization curves for flow cells in various environments. **e** Continuous electrolytic stability. The mass of the MoO$_3$@CoO/CC and Pt/C catalyst is about 1.89 and 1.5 mg cm$^{-2}$, respectively. **f** FE of producing H$_2$ and O$_2$, respectively. **g** Compared with the literature, the efficiency of producing H$_2$. See Supplementary Table 7 for details. The pH value of real seawater is about 8.0.

active interface of the catalyst, thus realizing selective catalysis; (2) the ultra-thin MoO$_3$ layer regulates the interface to obtain a high-activity catalytic site; (3) the CoMo-LDH phase limits the deep interface reconstruction, thereby reducing the corrosion of the catalyst. More importantly, the assembled MoO$_3$@CoO/CC||Pt/C/CC cell achieves an H$_2$ production rate of about 419.4 mL cm$^{-2}$ h$^{-1}$ for direct seawater electrolysis, and the corresponding power consumption is only 4.62 KWh/m$^3$ H$_2$. These results exceed those reported in real seawater electrolysis. This discovery not only develops a robust and stable catalyst to utilize abundant seawater sources for large-scale hydrogen production but also encloses a restricted dynamic surface reconstruction mechanism for guiding the design of high-performance OER catalysts.

## Methods
### Materials
Cobaltous nitrate hexahydrate (Co(NO$_3$)$_2$·6H$_2$O, 99%), ammonium fluoride (NH$_4$F, 99%), urea (CH$_4$N$_2$O, 99%), and ALD Mo precursor (Mo(CO)$_6$, 99.9%) was purchased commercially from Aladdin Company and used without further purification. The purity of oxygen and

nitrogen gas is 99.999%. All ALD processes were carried out in a commercial Sentech SI500 plasma-enhanced ALD reactor.

### Preparation of CoO/CC
Carbon fiber cloth (CC) was cleaned with acetone, ethanol, and deionized water and then dried in a 60 °C oven for 24 h. Co(NO$_3$)$_2$·6H$_2$O (291 mg), NH$_4$F (37 mg), and CH$_4$N$_2$O (300 mg) were added into 30 mL water to form a uniform solution, and a piece of CC (1 × 4 cm$^2$) was added for ultrasonic treatment for 1 h. Transfer it into a 50 mL stainless steel autoclave and keep it at 120 °C for 6 h to obtain the precursor of Co(OH)F nanowire array grown on CC (denoted as Co(OH)F/CC). Finally, Co(OH)F/CC was annealed at 700 °C for 2 h in an argon atmosphere to form beaded CoO nanostructures (denoted as CoO/CC). Refer to note S1 for other sample preparation.

### Preparation of MoO$_3$@CoO/CC
The amorphous MoO$_3$ layer was grown in ALD mode on the prepared CoO/CC substrate with alternating pulses of Mo(CO)$_6$ and oxygen plasma at 150 °C using the Sentech SI500 plasma-enhanced ALD instrument. Specific operation: 200 W RF power is used to generate O$_2$

plasma. One ALD cycle consists of a $Mo(CO)_6$ pulse for 3 s, an $N_2$ purge for 5 s, an $O_2$ plasma for 5 s, and an $N_2$ purge for 10 s. $MoO_3$ with different thicknesses can be obtained by controlling the number of deposition cycles (the thickness of $MoO_3$ deposited after 500 cycles is about 5 nm, named $MoO_3@CoO/CC$).

## Data availability

All relevant data that support the findings of this study are presented in the article and Supplementary Information. Source data are provided in this paper.

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

## Acknowledgements

This research was financially sponsored by the National Natural Science Foundation of China (52202109. D.G.; 52271225. X.C.; 52072273. S.W.; and 52331009. S.W.), Basic Scientific Research Projects of Wenzhou City (G20220024. D.G.; and H20220002. X.C.).

## Author contributions

D.G., X.C., and S.W. conceived the idea and directed the study. L.Z. and D.G. performed the main experiments. L.W. and Z.G. repeated the experiments and data. C.Z., G.F., and H.J. completed the computational studies. D.G., X.C., and S.W. drafted the manuscript. L.Z. participated in the experiments for sample preparation and characterization. All authors participated in the interpretation of the data and production of the final paper.

## Competing interests

The authors declare no competing interests.
