## [Peer Review File · Nature Communications]

A restricted dynamic surface self-reconstruction toward high-performance of direct seawater oxidationREVIEWER COMMENTS

Reviewer #1 (Remarks to the Author):

This manuscript designed a cowpea-shaped electrocatalyst ($\text{MoO}_3@\text{CoO}/\text{CC}$) for direct seawater oxidation. The authors claim that the ultra-thin amorphous MoO_3 could inhibit the phase segregation of Co-Mo bimetallic layered double hydroxide formed by dynamic self-reconfiguration of the catalyst interface, thereby improving the service life of the catalyst. In addition, the MoO_3 layer was also used to shield chloride ions from reaching the catalytic active interface, and the reconstructed stable CoMo LDH layer had electrostatic repulsion to further hydrophobic chlorine. Although the performance for direct seawater oxidation is good, the characterizations and analysis, as well as some key issues, are inaccurate or remain confusing. Considering the high standard of Nature Communications, I would like not to recommend its publication. The detailed comments are as follow.

1. It is not rigorous to judge the thickness of MoO_3 layer only by TEM. In addition, what is the point of the defects mentioned for several times? How does it function?
2. The shift of the XRD peak in Figure 2a appears to be an entire shift, which should be reprocessed.
3. The authors should measure the actual amount of each component.
4. The scan speed of the LSV curve is recommended to be 2 mV/s. Is there an iR-compensation during testing?
5. What is the active site of water oxidation reaction? What is the role of CoO component?
6. In Figure 3g for the stability test with a current density of 100mA cm^2 , the fluctuation of the potential is much smaller than the stability test with a lower current density, and even the voltage decreases within the first 40h, which is somewhat contrary to our knowledge.
7. Can Co and Mo form LDH structure?
8. Figure S14b shows that the XPS signal of Mo is significantly reduced after the reaction. Does this represent the dissolution of Mo?
9. In Figure 6i, why does the peak of OOH appear at only 1.1 V and disappear rapidly thereafter?
10. It is necessary to reprocess the XAS data carefully.
11. AEM electrolyzer is usually used to test real seawater splitting. Why do the authors use PEM electrolyzer?
12. One of the main challenges of seawater electrolysis, which the authors seem not to discuss, is the deposition of metal cations at the cathode. Is there any phenomenon in this aspect during the test of overall seawater splitting? The authors are necessary to explore it in detail.
13. It is weird that in Figures 7b and 7e, the control system ran with the presence with KOH but there is no KOH in Real Seawater system. If so, the control should run with only NaCl, which occupies a large amount in real seawater. Thus, the authors should re-do the performance tests in whole manuscript.

Reviewer #2 (Remarks to the Author):

In this study, Wang et al. presented a seawater oxidation electrocatalyst that achieves long-term stability

for more than 1000 h at 600 mA/cm²@η600 with high selectivity. As demonstrated, a new restricted dynamic surface self-reconstruction mechanism is induced by the formation a stable reconstructed Co-Mo double hydroxide phase interface layer. However, further experimental and simulation results need to be discussed, such as the theoretical calculation and mechanism discussion.

1. The natural seawater used in this manuscript has been alkalized, so the word "direct" is not appropriate.
2. What does "phase segregation of Co-Mo bimetallic" mean? (Line 79)
3. How did the authors determine the computational model? Any evidence from experiments? Why not use cobalt hydroxide and other phases with higher intrinsic activity for modeling and calculation? As shown in Figure 6j, directional reconstruction is emphasized. And the performance of the catalyst is obviously improved after reconstruction (Figure 5c). Furthermore, computational modeling is also difficult to correspond to the mechanism illustrated in Fig. S25. So I think the author's computational modeling would be problematic.
4. Although it is in strong alkali condition, the author uses PEM electrolyzer, which is usually used in pure water or weak acid environment. According to existing articles, anion exchange membrane (AEM) is used in most alkaline electrolyzers. In addition, the author can complete the specific treatment methods of proton exchange membrane for readers.
5. The contents of reconstruction in this manuscript should be discussed together, including HRTEM, in-situ Raman, and reconstruction mechanism diagram, etc.
6. The abscissa difference of CV curve is generally 0.1 V. (Figure 5c, S6, S7, S13)
7. It is obviously unreasonable to infer the mechanism of OER from EPR intensity, and it is suggested to prove the transformation of OER mechanism by other characterization methods, such as the in situ differential electrochemical mass spectrometry (DEMS) and pH-dependent experiment.
8. Please explain "Compared with Cl⁻, OH⁻ with smaller radius easily passes through MoO₃ layer." and "The CoMo LDH layered ...which has strong electrostatic repulsion to Cl⁻" in detail. Does the anti-chlorine corrosion effect of MoO₃ layer come from its passivation film or electrostatic attraction? In addition, does MoO₃ lead to the formation of MoO₄²⁻ under alkaline conditions? (Lines 222 and 281)

Reviewer #3 (Remarks to the Author):

The author synthesized ultra-thin amorphous MoO₃ using ALD technology on the surface of beaded CoO. At the MoO₃ and CoO interface, a new phase, identified as CoMo LDH after OER, demonstrated remarkable efficacy in hindering extensive interphase reconstruction, thereby enhancing the protection of genuine active sites. However, the current characterization and computational analysis mechanisms lack adequacy in substantiating this concept, making it challenging to endorse in esteemed journals like Nature Communications. The author should carefully address the following suggestions:

1. The identification of amorphous MoO₃ remains unclear. Firstly, the author did not synthesize crystalline MoO₃ using similar methods. Secondly, there is a lack of related characterization to systematically identify MoO₃.

2. In Fig. 2e, the author associates peaks at 531.48 and 532.51 eV with surface chemically adsorbed water for MoO₃@Co/CC and MoO₃/CC. However, a small peak near 532.51 eV is observed for both MoO₃@CoO/CC and CoO/CC. Confirm the accuracy of the XPS analysis.

3. In the DFT section, unify the notation for MoO₃@CoO/CC and MoO₃@CoO.

4. Figure 4 explores the migration energy barrier of H₂O/OH and Cl⁻ for MoO₃@CoO/CC and CoO/CC. It is crucial to supplement the associated computation about pure MoO₃, as the ionic migration path for MoO₃@CoO/CC is within MoO₃ in this article.

5. Although the article describes MoO₃ as amorphous, the DFT model depicts periodic MoO₃ in MoO₃@CoO/CC. Explain the correction of the model for the DFT calculation.

6. Corroborate the accuracy of the experimental model by fitting the DFT model with EXAFS data (Small 2021, 17, 2101163, DOI: 10.1002/smll.202101163).

7. In Fig. 5b, the author identifies darker areas as CoMo LDH based on lattice spacing. This characterization is insufficient; additional techniques such as EELs should be implemented.

8. Explain how the author identified CoMo LDH as an octahedral structure.

9. In Fig. 6, the article discusses the adsorbate evolution mechanism (AEM) and lattice-oxygen-mediated mechanism (LOM). Experimental verification is lacking; provide more experimental information following the reference (Adv. Mater. 2023, 2310690, DOI: 10.1002/adma.202310690).

Point-by-Point Responses to Reviewers' Comments

We are truly grateful to the reviewer's valuable comments and suggestions on our work, which has greatly improved the quality and clarity of this manuscript. All comments and suggestions have been taken into account in the revised manuscript, as described below.

Reviewer #1: This manuscript designed a cowpea-shaped electrocatalyst ($\text{MoO}_3@\text{CoO}/\text{CC}$) for direct seawater oxidation. The authors claim that the ultra-thin amorphous MoO_3 could inhibit the phase segregation of Co-Mo bimetallic layered double hydroxide formed by dynamic self-reconfiguration of the catalyst interface, thereby improving the service life of the catalyst. In addition, the MoO_3 layer was also used to shield chloride ions from reaching the catalytic active interface, and the reconstructed stable CoMo LDH layer had electrostatic repulsion to further hydrophobic chlorine. Although the performance for direct seawater oxidation is good, the characterizations and analysis, as well as some key issues, are inaccurate or remain confusing. Considering the high standard of Nature Communications, I would like not to recommend its publication. The detailed comments are as follow.

Author's response: We thank the reviewer for the importance comments of our work. We have carried out supplementary experiments and have made modifications according to the reviewer's constructive and valuable suggestions.

Comments #1: It is not rigorous to judge the thickness of MoO_3 layer only by TEM. In addition, what is the point of the defects mentioned for several times? How does it function?

Response: We truly appreciate the reviewer's helpful suggestions. In the manuscript, we firstly used SEM to analyze (see **Figure R1**) the diameter changes before and after the deposition of MoO_3 on CoO/CC surface, and counted a certain number of materials to get that the thickness of the grown MoO_3 is about 5 nm (See the inset in Fig. 1d and

Fig. S2b of the original manuscript). This result is consistent with TEM analysis (Fig. 1e). The corresponding modifications have been added to the manuscript.

In addition, ALD is based on a reaction between two or more gaseous precursors, which are pulsed alternately to avoid the presence of gas phase reactions. In this manner, the reactants are kept separated and react with surface functional groups. The process is self-limiting. Unreacted precursors and byproducts are removed by a purge step. The sequence of surface reactions and purges constitute an ALD cycle. Through this repetitive process, the film thickness can be accurately controlled.^[1, 2] However, it is difficult to measure the deposition rate of MoO₃ on the beaded-like CoO/CC surface. For this purpose, we deposited MoO₃ on a cobalt plate (containing a thin film of CoO on the surface) in order to test the growth rate. **Figure R2** shows that the growth of MoO₃ films follows the ideal saturation self-limiting ALD growth behavior at 150 °C, while the lengths of Mo(CO)₆ and O₂ plasma were maintained at 1 s and 4 s, respectively. The saturation growth rate of the ALD cycles was 0.0101 nm/cycles, and the film thickness was a linear relationship of the ALD cycle. Thus, after 500 cycles of deposition, the thickness of molybdenum oxide is 5 nm.

The significance of the deficiencies is mainly attributed to the following:

(1) Based on the self-limiting reaction characteristics of ALD. The precursors involved in the reaction during ALD will preferentially react with reactive groups (e.g., -OH, -NH₂, etc.)^[3, 4] and defective sites (vacancies, dangling bonds, cracks, grain boundaries, etc.)^[3] on the substrate in a surface chemical reaction. These sites enable precursors preferential adsorptions. Therefore, the ALD precursor chemically reacts preferentially with active groups and defects.^[3-5]

(2) Typically, defects are capable of altering the charge state of the catalyst surface, modulating the free energy of adsorption of key intermediates, and reducing the band gap (See Fig.2f and Fig.6c in the manuscript). Therefore, intrinsic defect sites intentionally introduced in a catalyst can be directly used as potential active sites.^[6-8]

Figure R1. SEM images of CoO/CC and MoO₃@CoO/CC, respectively. The illustration is the diameter statistics of various samples.

Figure R2. (a) Growth rate versus the O₂ plasma pulse length at fixed Mo precursor pulse length. (b) Growth rate versus the Mo precursor pulse length at fixed O₂ plasma pulse length. (c) Growth rate versus deposition temperature and (d) Film thickness as a function of total ALD cycles. The film thickness was measured by ellipsometer.

References:

[1] Wang, G. et al., Size-Selective Catalytic Growth of Nearly 100% Pure Carbon Nanocoils with Copper Nanoparticles Produced by Atomic Layer Deposition. *ACS Nano* **8**, 5330–5338 (2014).
 [2] Gao, Z. et al., Design and Properties of Confined Nanocatalysts by Atomic Layer Deposition. *Acc. Chem. Res.* **50**, 2309-2316 (2017).
 [3] Cao, K. et al., Inherently Selective Atomic Layer Deposition and Applications. *Chem. Mater.* **32**, 2195-2207 (2020).
 [4] Soethoudt, J. et al., Insight into Selective Surface Reactions of Dimethylamino-trimethylsilane

for Area-Selective Deposition of Metal, Nitride, and Oxide. *J. Phys. Chem. C* **124**, 7163-7173 (2020).

- [5] Yan, H. *et al.*, Single-Atom Pd(1)/Graphene Catalyst Achieved by Atomic Layer Deposition: Remarkable Performance in Selective Hydrogenation of 1,3-Butadiene. *J. Am. Chem. Soc.* **137**, 10484-10487 (2015).
- [6] Shen, W. *et al.*, Defect engineering of layered double hydroxide nanosheets as inorganic photosensitizers for NIR-III photodynamic cancer therapy. *Nat. Commun.* **13**, 3384 (2022).
- [7] Guo, D. *et al.*, Strategic Atomic Layer Deposition and Electrospinning of Cobalt Sulfide/Nitride Composite as Efficient Bifunctional Electrocatalysts for Overall Water Splitting. *Small* **16**, 2002432 (2020).
- [8] Xu, J. *et al.*, Atomic-level polarization in electric fields of defects for electrocatalysis. *Nat. Commun.* **14**, 7849 (2023).

Comments #2: The shift of the XRD peak in Figure 2a appears to be an entire shift, which should be reprocessed.

Response: We thank the reviewer for the insightful question. To clarify this problem, we re-tested the corresponding samples as shown in **Figure R3**. The characteristic peaks attributed to the carbon cloth were not shifted around 25.5° .^[1] The peaks of the CoO were significantly shifted negatively after the deposition of MoO₃. The corresponding modifications have been added to the manuscript.

Figure R3. XRD patterns of various samples.

References:

- [1] Lin, Q. *et al.*, Tuning the Interface of Co_{1-x}S/Co(OH)F by Atomic Replacement Strategy toward

Comments #3: The authors should measure the actual amount of each component.

Response: Thanks to reviewer for the valuable suggestions. We have obtained the corresponding catalyst content based on inductively coupled plasma optical emission spectrometer (ICP-OES) experiments in the calculation of turnover frequency (TOF). Due to our carelessness they were omitted in the original manuscript. Thus, the relevant data have been added in the supporting information.

The contents of CoO and MoO₃ in MoO₃@CoO/CC has been calculated as 1.87 mg/cm² and 0.02 mg/cm², respectively, according to the concentrations of Co²⁺ and Mo⁶⁺ in the sample.

Comments #4: The scan speed of the LSV curve is recommended to be 2 mV/s. Is there an iR-compensation during testing?

Response: Thank you for your precious comments and advices. In the experiment, we explored the influence of LSV scanning speed on performance as shown in **Figure R4a**. Sweep speed from 2 to 50 mV s⁻¹ exhibits some effect on the performance. However, the influence is limited. Considering that the catalyst interface can quickly reach stability at a high scanning speed,^[1,2] we choose 20 mV s⁻¹ as scan speed in the subsequent experiment.

There is no iR-compensation during the experimental test. To clarify this problem, we compare the difference between iR-compensation and no compensation, as shown in **Figure R4b**. The results show that iR-compensation and no compensation have the similar initial over-potential. Considering the practical application,^[3] we choose uncompensated in the subsequent experiment.

Figure R4. (a) Linear sweep voltammetry (LSV) curves of MoO₃@CoO/CC at various scanning speeds. (b) Comparison of LSV curves of MoO₃@CoO/CC with iR-compensation and uncompensated in artificial seawater.

References:

- [1] Chauhan, M. et al., Copper Cobalt Sulfide Nanosheets Realizing a Promising Electrocatalytic Oxygen Evolution Reaction. *ACS Catal.* **7**, 5871–5879 (2017).
- [2] Guo, D. Y. et al., A CoN-based OER Electrocatalyst Capable in Neutral Medium: Atomic Layer Deposition as Rational Strategy for Fabrication. *Adv. Funct. Mater.* **31**, 2101324 (2021).
- [3] Son, Y. J. et al., Navigating iR Compensation: Practical Considerations for Accurate Study of Oxygen Evolution Catalytic Electrodes. *ACS Energy Lett.* **8**, 4323–4329 (2023).

Comments #5: What is the active site of water oxidation reaction? What is the role of CoO component?

Response: We thank the reviewer for the insightful question. The active site of water oxidation reaction is CoO interface modified by MoO₃. As shown in **Figure R5** (See manuscript Fig. 3a), the activity of MoO₃/CC catalyst is very weak, which is obviously lower than that of CoO/CC catalyst in water oxidation reaction. ALD MoO₃ is constructed on CoO/CC interface, which significantly improved the catalytic activity. Based on the above explanations, CoO is recognized as the catalytic active centre and MoO₃ is the co-catalyst.

Figure R5. Linear sweep voltammetry (LSV) curves of various samples.

Comments #6: In Figure 3g for the stability test with a current density of 100 mA cm^{-2} , the fluctuation of the potential is much smaller than the stability test with a lower current density, and even the voltage decreases within the first 40h, which is somewhat contrary to our knowledge.

Response: Thank you for your precious comments and advices. It should be noted that in the experiment, we used the same sample electrode to continuously test the stability under different current densities from small to large currents. Moreover, we have not been activated before testing. At 10 mA cm^{-2} , the interface of the catalyst has a stable process, and the interface change can be accurately captured at a small current.^[1] In addition, when starting electrolysis at a current of 50 mA cm^{-2} , we added new electrolyte to the electrolytic cell, which will lead to a relatively large fluctuation of stability in the early stage.

To clarify this problem, we re-tested the stability of $\text{MoO}_3@\text{CoO}/\text{CC}$ under different current densities by chronoamperometry method. We carefully excluded the interfering effects of the above factors, and the test results are shown in **Figure R6**. The results show that the voltage increases by 8, 20 and 35 mV after 100 h of continuous oxygen evolution at current densities of 10, 50 and 100 mA cm^{-2} , respectively, indicating that the stability decreases gradually with the increase of current density. As the reviewer said, the fluctuation of the potential is smaller with a lower current density than larger current for the stability test. The revised data is supplemented in the manuscript Fig. 3g.

Figure R6. Evaluation of MoO₃@CoO/CC stability by chronoamperometry method.

References:

[1] Guo, J. *et al.*, Direct seawater electrolysis by adjusting the local reaction environment of a catalyst. *Nat. Energy* **8**, 264–272 (2023).

Comments #7: Can Co and Mo form LDH structure?

Response: We thank the reviewer for the insightful question. The surface of OER catalyst tends to be self-reconstructed by oxidation at high anode potential to form hydrogen (hydroxyl) oxide.^[1-3] The structure of CoMo LDH was characterized by TEM in the experiment. A lattice spacing of 0.38 nm was observed between the MoO₃ layer and CoO (See Fig. 5b1 in the manuscript), corresponding to the (006) crystal plane of CoMo LDH.^[4] In addition, the corresponding XPS and XAS analyses demonstrate the formation of CoMo LDH during the OER process (See Fig.5e-i in the manuscripts).

More importantly, the OER performance of the MoO₃@CoO/CC catalyst gradually became better after different numbers of oxygen evolution (**Figure R7**), indicating that the catalyst interface was reconstructing.^[2,3] With subsequent linear scanning voltammetry cycles (**inset Figure R7**), the oxidation peak of Co²⁺ shifted to a higher potential, indicating that a portion of Mo ions initially in MoO₃ on the CoO surface dispersed into CoOOH to form CoMo LDH.^[5] In summary, the CoMo LDH phase could be formed between MoO₃ and CoO as the OER reaction proceeds.

Figure R7. Linear scanning voltammetric curves of MoO₃@CoO/CC catalyst after different oxygen evolution times.

References:

- [1] Chung, D. Y. *et al.*, Dynamic stability of active sites in hydr(oxy)oxides for the oxygen evolution reaction. *Nat. Energy* **5**, 222-230 (2020).
- [2] Kang, J. X. *et al.*, Valence oscillation and dynamic active sites in monolayer NiCo hydroxides for water oxidation. *Nat. Catal.* **4**, 1050-1058 (2021).
- [3] Ye, S. H. *et al.*, Deeply self-reconstructing CoFe(H₃O)(PO₄)₂ to low-crystalline Fe_{0.5}Co_{0.5}OOH with Fe³⁺-O-Fe³⁺ motifs for oxygen evolution reaction. *Appl. Catal. B: Environ.* **304**, 120986 (2022).
- [4] Shen, W. *et al.*, Defect engineering of layered double hydroxide nanosheets as inorganic photosensitizers for NIR-III photodynamic cancer therapy. *Nat. Commun.* **13**, 3384 (2022).
- [5] Ou, Y. Q. *et al.*, Cooperative Fe sites on transition metal (oxy)hydroxides drive high oxygen evolution activity in base. *Nat. Commun.* **14**, 7688 (2023).

Comments #8: Figure S14b shows that the XPS signal of Mo is significantly reduced after the reaction. Does this represent the dissolution of Mo?

Response: We thank the reviewer for the insightful questions. As shown in **Figure R8**, the sample became coarse obviously and its diameter increased after the OER reaction. This is mainly attributed to the fact that after OER, the interface between molybdenum oxide layer and cobalt oxide is oxidized at high anode potential and self-reconstructed to form hydrogen (hydroxyl) oxide.^[1,2] Due to the limited depth of XPS test and the increase of surface oxygen content, the content of other elements such as Co and Mo will decrease (Table R1, See Fig.5d and Figure S15 in the

manuscript).

In addition, we tested the electrolyte after OER with ICP-OES and no Mo element was detected, indicating that Mo was not dissolved during OER.

Figure R8. SEM images of (a) MoO₃@CoO/CC and (b) MoO₃@CoO/CC-after. (c) High-resolution XPS spectra of Mo 3d before and after the oxygen evolution reaction of MoO₃@CoO/CC catalyst.

Table R1. Before and after OER, the contents of various elements in sample MoO₃@CoO/CC were tested by XPS.

Sample	C (Atomic %)	O (Atomic %)	Co (Atomic %)	Mo (Atomic %)
MoO ₃ @CoO/CC	39.98	41.97	10.31	7.74
MoO ₃ @CoO/CC-after	34.86	50.24	8.99	5.91

References:

- [1] Ye, S. H. *et al.*, Deeply self-reconstructing CoFe(H₃O)(PO₄)₂ to low-crystalline Fe_{0.5}Co_{0.5}OOH with Fe³⁺-O-Fe³⁺ motifs for oxygen evolution reaction. *Appl. Catal. B: Environ.* **304**, 120986 (2022).
- [2] Guo, D., Wan, Z., Li, Y., Xi, B. & Wang, C., TiN@Co_{5.47}N Composite Material Constructed by Atomic Layer Deposition as Reliable Electrocatalyst for Oxygen Evolution Reaction. *Adv. Funct. Mater.* **31**, 2008511 (2021).
- [3] Zhai, P. *et al.*, Regulating electronic states of nitride/hydroxide to accelerate kinetics for oxygen evolution at large current density. *Nat. Commun.* **14**, 1873 (2023).

Comments #9: In Figure 6i, why does the peak of OOH appear at only 1.1 V and disappear rapidly thereafter?

Response: Response: As shown in **Figure R9a**, CoO is oxidized to +3 at about 1.1 V to form the CoOOH phase, which is further oxidized to CoO₂ with the highest valence

at about 1.5V.^[1] For this reason, as the voltage increases, the surface is covered with more and more Co with the highest valence, which leads to the gradual weakening of the characteristic peak of -OOH (**Figure R9b**). This process is called as deep reconstruction, which leads to the phase separation of the catalyst.^[2]

Figure R9. (a) CV curves of CoO/CC. (b) In situ Raman spectrum of the CoO/CC in 1 M KOH+0.5 M NaCl measured at different potentials.

References:

- [1] Zhang, Y. *et al.*, Rapid synthesis of cobalt nitride nanowires: Highly efficient and low-cost catalysts for oxygen evolution. *Angew. Chem. Int. Ed.* **55**, 8670-8674 (2016).
- [2] Ye, S. H. *et al.*, Deeply self-reconstructing CoFe(H₃O)(PO₄)₂ to low-crystalline Fe_{0.5}Co_{0.5}OOH with Fe³⁺-O-Fe³⁺ motifs for oxygen evolution reaction. *Appl. Catal. B: Environ.* **304**, 120986 (2022).

Comments #10: It is necessary to reprocess the XAS data carefully.

Response: Thank you very much for your kind reminding for us. To clarify the above questions of reviewers. According to your comments, The XAS of various samples were re-fitted and the data were analyzed in detail as shown in the manuscript. The corresponding changes have been added to the manuscript and highlighted with a yellow background. Especially, the fitting of Mo-K edge is shown in **Figure 10R**. The fitting parameters of each sample are shown in **Table R2**, and the whole fitting ΔE value and R factor meet the requirements of fitting law.^[1-5]

Figure R10. k space fitting curves of (a, b) Mo foil, (c, d) MoO₃@CoO/CC, (e, f) MoO₃@CoO/CC-after, (g, h) MoO₃ and (i, j) MoCl₅ at Mo K-edge.

Table R2. EXAFS fitting parameters at the **M** *K*-edge (M=Co, Mo) for various samples.

Sample	Shell	CN^a	$R(\text{\AA})^b$	$\sigma^2(\text{\AA}^2)^c$	$\Delta E_0(\text{eV})^d$	R factor
Co K -edge ($S_0^2=0.753$)						
Co foil	Co-Co	12*	2.494±0.003	0.0061±0.0001	6.7±0.3	0.0029
CoO	Co-O	6.0±0.4	2.077±0.022	0.0102±0.0018	-2.1±0.9	0.0063
	Co-Co	11.8±0.5	3.016±0.005	0.0081±0.0008		
MoO ₃ @ CoO/CC	Co-O1	0.8±0.5	1.985±0.013	0.0082±0.0024	1.5±3.6	0.0081
	Co-O2	4.7±0.3	2.105±0.026			
	Co-Co	10.1±0.8	3.010±0.017	0.0094±0.0005	1.2±2.1	
	Co-O-Mo	1.2±0.3	3.198±0.024			
MoO ₃ @ CoO/CC- after	Co-O1	2.4±0.3	1.906±0.017	0.0052±0.0045	2.1±6.1	0.0137
	Co-O2	3.4±0.1	2.134±0.016			
	Co-Co	9.7±0.5	3.003±0.019	0.0105±0.0021	1.6±6.9	
	Co-O-Mo	0.8±0.3	3.213±0.016			
Mo K -edge ($S_0^2=0.803$)						
Mo foil	Mo-Mo	8*	2.721±0.005	0.0031±0.0008	-8.4±0.7	0.0005
	Mo-Mo	6*	3.124±0.003	0.0024±0.0008		
MoO ₃	Mo-O	6.0±0.5	1.956±0.005	0.0034±0.0005	-2.4±1.0	0.0078
	Mo-Mo	5.0±0.7	3.651±0.004	0.0032±0.0004		
MoO ₃ @ CoO/CC	Mo-O	4.8±0.5	1.846±0.011	0.0106±0.0046	-3.4±6.8	0.0086
	Mo-Mo	5.0±0.2	3.147±0.005	0.0023±0.0006		
	Mo-O-Co	2.3±0.3	1.78±0.011	2.6±0.6	-6.7±0.6	
MoO ₃ @ CoO/CC- after	Mo-O	4.5±0.2	2.03±0.005	0.0031±0.0014	2.9±1.4	0.0043
	Mo-Mo	4.2±0.2	2.52±0.002	0.0056±0.0003		
	Mo-O-Co	2.4±0.1	1.04±0.08	3.17±0.046	-3.1±0.6	
	Mo-Cl	2.8±0.1	1.76±0.09	2.25±0.038		
MoCl ₅	Mo-Mo	5.2±0.8	2.12±0.008	0.051±0.0007	-5.2±1.5	0.0025
	Mo-Cl	5.1±0.2	2.14±0.05	4.14±0.06		

^a*CN*, coordination number; ^b*R*, the distance to the neighboring atom; ^c σ^2 , the Mean Square Relative Displacement (MSRD); ^d ΔE_0 , inner potential correction; *R* factor indicates the goodness of the fit. S_0^2 was fixed to 0.753 and 0.803, according to the experimental EXAFS fit of Co foil and Mo foil by fixing *CN* as the known crystallographic value. * This value was fixed during EXAFS fitting, based on the known structure of Co and Mo. Fitting range: $3.0 \leq k (\text{\AA}^{-1}) \leq 14.0$ and $1.0 \leq R (\text{\AA}) \leq 3.0$ (Co foil); $2.0 \leq k (\text{\AA}^{-1}) \leq 13.6$ and $1.0 \leq R (\text{\AA}) \leq 3.5$ (MoO₃@CoO/CC); $2.0 \leq k (\text{\AA}^{-1}) \leq 13.5$ and $1.0 \leq R (\text{\AA}) \leq 3.5$ (MoO₃@CoO/CC-after); $3.0 \leq k (\text{\AA}^{-1}) \leq 12.0$ and $1.0 \leq R (\text{\AA}) \leq 3.0$ (Mo foil); $2.0 \leq k$

(\AA) ≤ 12.7 and $1.0 \leq R$ (\AA) ≤ 3.0 (MoO_3); $2.0 \leq k$ (\AA) ≤ 12.7 and $1.0 \leq R$ (\AA) ≤ 3.0 (MoCl_5). A reasonable range of EXAFS fitting parameters: $0.700 < S_0^2 < 1.000$; $CN > 0$; $\sigma^2 > 0 \text{\AA}^2$; $|\Delta E_0| < 10 \text{ eV}$; R factor < 0.02 .

References:

- [1] Ravel, B. et al., ATHENA, ARTEMIS, HEPHAESTUS: data analysis for X-ray absorption spectroscopy using IFEFFIT, *J. Synchrotron Radiat.* **12**, 537–541 (2005).
- [2] Funke, H. et al., Wavelet analysis of extended X-ray absorption fine structure data. *Phys. Rev. B* **71**, 094110 (2005).
- [3] Zabinsky, S. I. et al., Multiple-Scattering Calculations of X-Ray-Absorption Spectra. *Phys. Rev. B* **52**, 2995–3009 (1995).
- [4] Fei, H. et al., Atomic cobalt on nitrogen-doped graphene for hydrogen generation. *Nat. Commun.* **6**, 8668-8676 (2015).
- [5] Kuai, C. et al., Phase segregation reversibility in mixed-metal hydroxide water oxidation catalysts. *Nat. Catal.* **3**, 743-753 (2020).

Comments #11: AEM electrolyzer is usually used to test real seawater splitting. Why do the authors use PEM electrolyzer?

Response: Response: We are very grateful for your insightful questions. As the reviewer said, AEM electrolytic cell is generally selected for splitting seawater with weak alkalinity in real seawater.^[1] It is worth noting that in the manuscript we mention the use of flow cells using nafion as a separation (See the analysis of manuscripts Lines 419 and 421 for details). In the original manuscript, we chose an amphoteric nafion membrane, which is mainly because the $\text{MoO}_3@\text{CoO}/\text{CC}$ material we designed can effectively adsorb and shield metal cations in seawater (**Figure R11**), thus improving the service life of the catalyst in the whole electrode (For a detailed description, seeing the manuscript support information Fig. S31). In other words, the separator in this paper only plays a role in inhibiting the opposite diffusion of gas. Unfortunately, we didn't carefully check this part of the description in the supporting information. The corresponding modification has been added to the manuscript.

In order to clarify this problem, we present data from a flow electrolytic cell test using an anion-exchange membrane Tokuyama A201, Pt/C, and $\text{MoO}_3@\text{CoO}/\text{CC}$ as the separator, cathode, and anode, respectively. As shown in **Figure R12**, the curves of

both diaphragm-assembled electrolytic cells tested at an industrial hydrogen production current density of 1.0 A cm^{-2} for 100 h showed good stability. The above results further show that our designed $\text{MoO}_3@\text{CoO}/\text{CC}$ exhibits good performance of direct electrolysis of seawater.

Figure R11. (a) Schematic diagram and optical photograph of flow electrolytic cell. (b) Schematic diagram of seawater in and out of the flow electrolytic cell and the effect of $\text{MoO}_3@\text{CoO}/\text{CC}$ on seawater.

Figure R12. The stability test of flow electrolytic cell using two kinds of separator assembly.

References:

[1] Dresch, S. *et al.*, Efficient direct seawater electrolyzers using selective alkaline NiFe-LDH as OER catalyst in asymmetric electrolyte feeds. *Energy Environ. Sci.* **13**, 1725-1729 (2020).

Comments #12: One of the main challenges of seawater electrolysis, which the authors seem not to discuss, is the deposition of metal cations at the cathode. Is there any phenomenon in this aspect during the test of overall seawater splitting? The authors are necessary to explore it in detail.

Response: Thanks for the reviewer's comments. Seawater electrolysis can be divided into cathodic hydrogen evolution reaction (HER) and anodic oxygen evolution reaction (OER), accompanied by the competition of halides in anodic chlorine evolution reaction (CER).^[1,2]

The main problems in cathodic HER are the presence of various soluble cations (Na^+ , Mg^{2+} , Ca^{2+} , etc.), bacteria/microorganisms and solid impurities/precipitates that may poison the catalyst, electrodes and membranes and shorten the stable life of the electrolyzer.^[3,4]

The problem existing in that anode are as follow: First, CER is competitive to OER at anode that lowers the OER selectivity and forms toxic chlorine.^[3] Second, the strong binding energy between Cl^- and active sites of the electrocatalysts accelerates catalyst corrosion and leads to poor durability. Electrolysis at high current densities is crucial for practical applications, but the above problems become more serious than that at low current densities ($<200 \text{ mA cm}^{-2}$).^[5] As a result, the highest current densities delivered constantly by most of the seawater electrocatalysts reported so far remain below the industrial requirements of 500 mA cm^{-2} ,^[6] and it is rare that the electrocatalysts work stably for over 200 h.^[7] Therefore, the main challenge of seawater electrolysis is how to prevent CER from selectively performing OER at anode.^[8] The other major challenge of seawater splitting is how to control the process of catalyst reconstruction to avoid phase separation caused by deep reconstruction, which becomes the key challenge to obtain high activity OER catalyst.^[9-11] For this reason, this paper focuses on the key scientific question of how to avoid deep catalyst reconfiguration and selective seawater oxidation. The designed $\text{MoO}_3@\text{CoO}/\text{CC}$ catalysts were verified by a series of experiments to be capable of restricted dynamic surface self-reconstruction and selective OER during electrolytic seawater oxidation, and the related mechanisms

were revealed in detail. Finally, we further evaluated the ability to design catalysts for the direct electrolysis of seawater by forming the anode and cathode of a flow electrolysis cell with MoO₃@CoO/CC and Pt/C, respectively. For this reason, we don't discuss the problems of cathode in electrolytic seawater too much in this paper.

It is worth noting that in a flow cell seawater enters from the anode side and passes through the exchange film to the cathode side (see Supplementary Fig. 31). Due to the anode is made of MoO₃@CoO/CC material, it has high selectivity for Cl⁻ and can effectively inhibit CER, thus realizing direct seawater oxidation. In addition, because MoO₃ has an octahedral structure of MoO₆, oxygen atoms are placed at the top of the octahedron, which have a good adsorption capacity for seawater cations, thus avoiding the cations from reaching the cathode and improving the catalytic stability of the Pt/C/CC electrode. As shown in Supplementary Fig. 30, the RuO₂/CC (+) || Pt/C/CC electrolytic cell shows poor stability and Faraday efficiency, indicating that cations and anions in seawater have great influence on the catalyst.

To clarify the reviewers' questions, we dissolved the corresponding cathode Pt/C of MoO₃@CoO/CC||Pt/C/CC and RuO₂/CC||Pt/C/CC electrolytic cells after continuous electrolysis for 300 h in aqua regia solution to test ICP-OES, respectively. The results shows that the concentrations of Ca²⁺ and Mg²⁺ ions examined at the cathode of the MoO₃@CoO/CC||Pt/C/CC electrolysis cell are 0.57% and 0.88%, respectively. The concentrations of Ca²⁺ and Mg²⁺ ions were examined in the cathode of RuO₂/CC||Pt/C/CC electrolytic cell are 35.2% and 37.3% mmol/L, respectively. The above analysis results further show that MoO₃@CoO/CC catalyst exhibits good direct seawater oxidation ability, and can inhibit the cations in seawater from reaching the cathode in the flow electrolytic cell, thus protecting the cathode catalyst.

References:

- [1] He, W. J. et al., Materials Design and System Innovation for Direct and Indirect Seawater Electrolysis. *ACS Nano* **17**, 22227-22239 (2023).
- [2] Guo, J. et al., Direct seawater electrolysis by adjusting the local reaction environment of a catalyst. *Nat. Energy* **8**, 264–272 (2023).
- [3] Xie, H. P. et al., A membrane-based seawater electrolyser for hydrogen generations, *Nature* **612**, 673-678 (2022).

- [4] Shi, L. *et al.*, Using reverse osmosis membranes to control ion transport during water electrolysis. *Energy Environ. Sci.* **13**, 3138-3148 (2020).
- [5] Luo, Y. *et al.* Morphology and surface chemistry engineering toward pH-universal catalysts for hydrogen evolution at high current density. *Nat. Commun.* **10**, 269 (2019).
- [6] Vos, J. G., Wezendonk, T. A., Jeremiase, A. W. & Koper, M. T. M. MnO_x/IrO_x as selective oxygen evolution electrocatalyst in acidic chloride solution. *J. Am. Chem. Soc.* **140**, 10270–10281 (2018).
- [7] Wu, B. *et al.* A unique NiOOH@FeOOH heteroarchitecture for enhanced oxygen evolution in saline water. *Adv. Mater.* **34**, 2108619 (2022).
- [8] Kang, X. *et al.*, A corrosion-resistant RuMoNi catalyst for efficient and long-lasting seawater oxidation and anion exchange membrane electrolyzer. *Nat. Commun.* **14**, 3607 (2023).
- [9] Wang, W., Duan, J., Liu, Y. & Zhai, T., Structural Reconstruction of Catalysts in Electroreduction Reaction: Identifying, Understanding, and Manipulating. *Adv. Mater.* **34**, 2110699 (2022).
- [10] Karlsson, R. K. B. & Cornell, A., Selectivity between Oxygen and Chlorine Evolution in the Chlor-Alkali and Chlorate Processes. *Chem. Rev.* **116**, 2982-3028 (2016).
- [11] Kuai, C. *et al.*, Phase segregation reversibility in mixed-metal hydroxide water oxidation catalysts. *Nat. Catal.* **3**, 743-753 (2020).

Comments #13: It is weird that in Figures 7b and 7e, the control system ran with the presence with KOH but there is no KOH in Real Seawater system. If so, the control should run with only NaCl, which occupies a large amount in real seawater. Thus, the authors should re-do the performance tests in whole manuscript.

Response: Thanks for the reviewer's comments. The pH value of real seawater is around 8.0, which is a weak alkaline environment.^[1,2,3] The pure NaCl solution is under a neutral condition. Moreover, with the progress of electrolysis, the pH value gradually increased.^[1-9] At present, researchers mostly use alkaline artificial seawater to simulate real seawater.^[5-9] Besides, the concentration of salt will gradually increase with the continuous electrolysis of seawater in practical electrolysis applications.^[10] For this reason, the research concept of this paper is to firstly investigate the electrical properties of MoO₃@CoO/CC catalysts under artificial alkaline seawater, and then to further investigate the applicability of the catalysts by increasing the alkaline and salt concentrations as well as under real seawater.

References:

- [1] Guo, J. *et al.*, Direct seawater electrolysis by adjusting the local reaction environment of a

- catalyst. *Nat. Energy* **8**, 264–272 (2023).
- [2] He, W. J. et al., Materials Design and System Innovation for Direct and Indirect Seawater Electrolysis. *ACS Nano* **17**, 22227-22239 (2023).
- [3] Xie, H. P. et al., A membrane-based seawater electrolyser for hydrogen generations, *Nature* **612**, 673-678 (2022).
- [4] Luo, Y. et al. Morphology and surface chemistry engineering toward pH-universal catalysts for hydrogen evolution at high current density. *Nat. Commun.* **10**, 269 (2019).
- [5] Wu, B. et al. A unique NiOOH@FeOOH heteroarchitecture for enhanced oxygen evolution in saline water. *Adv. Mater.* **34**, 2108619 (2022).
- [6] Yu, L. et al., Ultrafast room-temperature synthesis of porous S-doped Ni/Fe (oxy)hydroxide electrodes for oxygen evolution catalysis in seawater splitting. *Energy Environ. Sci.* **13**, 3439-3446 (2020).
- [7] Yu, L. et al., High-performance seawater oxidation by a homogeneous multimetallic layered double hydroxide electrocatalyst. *PNSA* **119**, e2202382119 (2022).
- [8] Xu, W. W. et al., Ag Nanoparticle-induced Surface Chloride Immobilization Strategy enables Stable Seawater Electrolysis. *Adv. Mater.* **35** 2306062 (2023).
- [9] Kang, X. et al., A corrosion-resistant RuMoNi catalyst for efficient and long-lasting seawater oxidation and anion exchange membrane electrolyzer. *Nat. Commun.* **14**, 3607 (2023).
- [10] Kuang, Y. et al., Solar-driven, highly sustained splitting of seawater into hydrogen and oxygen fuels. *PNAS.* **116**, 6624-6629 (2019).

Reviewer 2: In this study, Wang et al. presented a seawater oxidation electrocatalyst that achieves long-term stability for more than 1000 h at 600 mA/cm²@η600 with high selectivity. As demonstrated, a new restricted dynamic surface self-reconstruction mechanism is induced by the formation a stable reconstructed Co-Mo double hydroxide phase interface layer. However, further experimental and simulation results need to be discussed, such as the theoretical calculation and mechanism discussion.

Author's response: We thank the reviewer for acknowledging the importance of our work. We have carried out supplementary experiments and have made modifications according to the reviewer's constructive and valuable suggestions.

Comments #1: The natural seawater used in this manuscript has been alkalized, so the word "direct" is not appropriate.

Response: Thanks for the reviewer's comments. The pH value of real seawater is around 8.0, which is a weak alkaline environment.^[1,2,3] Besides, with the progress of electrolysis, the pH value gradually increased.^[1-9] At present, researchers mostly use

alkaline artificial seawater to simulate real seawater.^[5-9] Besides, the concentration of salt will gradually increase with the continuous electrolysis of seawater in practical electrolysis applications.^[10] For this reason, the research concept of this paper is to firstly investigate the electrical properties of MoO₃@CoO/CC catalysts under artificial alkaline seawater, and then to further investigate the applicability of the catalysts by increasing the alkaline and salt concentrations as well as under real seawater. For this reason, most researchers often use this idea to study the direct electrolysis of seawater.^[1-9]

References:

- [1] Guo, J. *et al.*, Direct seawater electrolysis by adjusting the local reaction environment of a catalyst. *Nat. Energy* **8**, 264–272 (2023).
- [2] He, W. J. *et al.*, Materials Design and System Innovation for Direct and Indirect Seawater Electrolysis. *ACS Nano* **17**, 22227-22239 (2023).
- [3] Xie, H. P. *et al.*, A membrane-based seawater electrolyser for hydrogen generations, *Nature* **612**, 673-678 (2022).
- [4] Luo, Y. *et al.* Morphology and surface chemistry engineering toward pH-universal catalysts for hydrogen evolution at high current density. *Nat. Commun.* **10**, 269 (2019).
- [5] Wu, B. *et al.* A unique NiOOH@FeOOH heteroarchitecture for enhanced oxygen evolution in saline water. *Adv. Mater.* **34**, 2108619 (2022).
- [6] Yu, L. *et al.*, Ultrafast room-temperature synthesis of porous S-doped Ni/Fe (oxy)hydroxide electrodes for oxygen evolution catalysis in seawater splitting. *Energy Environ. Sci.* **13**, 3439-3446 (2020).
- [7] Yu, L. *et al.*, High-performance seawater oxidation by a homogeneous multimetallic layered double hydroxide electrocatalyst. *PNSA* **119**, e2202382119 (2022).
- [8] Xu, W. W. *et al.*, Ag Nanoparticle-induced Surface Chloride Immobilization Strategy enables Stable Seawater Electrolysis. *Adv. Mater.* **35** 2306062 (2023).
- [9] Kang, X. *et al.*, A corrosion-resistant RuMoNi catalyst for efficient and long-lasting seawater oxidation and anion exchange membrane electrolyzer. *Nat. Commun.* **14**, 3607 (2023).
- [10] Kuang, Y. *et al.*, Solar-driven, highly sustained splitting of seawater into hydrogen and oxygen fuels. *PNAS.* **116**, 6624-6629 (2019).

Comments #2: What does "phase segregation of Co-Mo bimetallic" mean? (Line 79)

Response: We thank the reviewer for the insightful question. The catalysts are prone to initiate the self-restructuring reaction between the catalysts and high-activity nascent intermediates (e.g., O*, HO* and HOO*) under the OER high overpotential, thus forming hydroxyl metal oxides.^[1-3] Unrestricted reconfiguration causes deep catalyst restructuring, which results in the formation of bimetallic layered double hydroxides

(M₁M₂ LDH) from the pre-reconstruction, gradual segregation into monometallic hydroxyl oxides (M₁OOH/M₂OOH), and further formation of higher valence oxides.^[4-6] This process is called phase separation.^[7] The formation of M₁M₂ LDH is considered as one of the active components of the catalyst.^[1-6] However, the phase separation process leads to catalyst deactivation and for this reason this process is not desired.^[3-6]

References:

- [1] Guo, J. *et al.*, Direct seawater electrolysis by adjusting the local reaction environment of a catalyst. *Nat. Energy* **8**, 264–272 (2023).
- [2] Kang, J. X. *et al.*, Valence oscillation and dynamic active sites in monolayer NiCo hydroxides for water oxidation. *Nat. Catal.* **4**, 1050-1058 (2021).
- [3] Ye, S. H. *et al.*, Deeply self-reconstructing CoFe(H₃O)(PO₄)₂ to low-crystalline Fe_{0.5}Co_{0.5}OOH with Fe³⁺-O-Fe³⁺ motifs for oxygen evolution reaction. *Appl. Catal. B: Environ.* **304**, 120986 (2022).
- [4] Chen, F. Y. *et al.*, Stability challenges of electrocatalytic oxygen evolution reaction: From mechanistic understanding to reactor design. *Joule* **5**, 1704-1731 (2021).
- [5] Peng, L. *et al.*, Atomic Cation-Vacancy Engineering of NiFe-Layered Double Hydroxides for Improved Activity and Stability towards the Oxygen Evolution Reaction. *Angew. Chem. Int. Ed.* **60**, 24612-24619 (2021).
- [6] Shi, Z. *et al.*, Confined Ir single sites with triggered lattice oxygen redox: Toward boosted and sustained water oxidation catalysis. *Joule* **5**, 2164-2176 (2021).
- [7] Kuai, C. *et al.*, Phase segregation reversibility in mixed-metal hydroxide water oxidation catalysts. *Nat. Catal.* **3**, 743-753 (2020).

Comments #3: How did the authors determine the computational model? Any evidence from experiments? Why not use cobalt hydroxide and other phases with higher intrinsic activity for modeling and calculation? As shown in Figure 6j, directional reconstruction is emphasized. And the performance of the catalyst is obviously improved after reconstruction (Figure 5c). Furthermore, computational modeling is also difficult to correspond to the mechanism illustrated in Fig. S25. So I think the author's computational modeling would be problematic.

Response: Thank you for your precious comments and advices. The model in this paper is mainly based on the XRD of the prepared samples. In Fig. 2a, the X-ray diffraction (XRD) pattern of beaded-like CoO/CC is very consistent with the standard peak of CoO cubic crystal phase (JCPDS No.48–1719).^[1]

The amorphous MoO₃ structure constructed by ALD mainly determines the model through the following experiments: From the XPS spectrum of MoO_x thin films prepared by ALD, it can be seen that they mainly contain Mo, C and O elements (**Figure R13a**). The contents of various elements analyzed by XPS are shown in **Figure R13b**. The results show that the atomic contents of Mo and O in the prepared MoO_x are close to 1:3, indicating that the phase of the prepared material is MoO₃. In **Figure R13c**, the XPS spectrum shows double peaks at 232.7 and 235.8 eV, corresponding to Mo 3d^{5/2} and Mo 3d^{3/2},^[2,3] respectively. These two peaks belong to the completely oxidized Mo⁶⁺ state, corresponding to the stoichiometric structure of MoO₃. It can be seen that the wavelength at 302, 433, 619, 670, 824, 963 cm⁻¹ corresponds to the Mo-O bond in MoO_x (**Figure R13d**), which indicates the presence of MoO₃ species.^[4,5] From **Figure R13e** that different annealing temperatures have great influence on the crystallization of the film. When the annealing temperature reaches 400 °C, α-MoO₃ diffraction peaks appear at 12.8°, 25.7° and 39.1°, and β-MoO₃ diffraction peaks appear at 23° and 25°.^[6,7] With the temperature rising to 500 °C, the XRD peak of β-MoO₃ disappeared, indicating that the purity of the film is higher.^[7] The above analysis verified that amorphous MoO₃ was prepared by ALD.

The structure of MoO₃@CoO/CC is mainly determined by the following experiments: The strong electronic interaction between CoO and MoO₃ is verified in detail in the "Structural Analysis" section of the paper (See Fig.2 in the manuscript). In addition, MoO₃ is verified by XAS to be anchored to the CoO surface with Mo-O-Co bonds (See Fig.5e-i in the manuscript). The above analysis is the experimental evidence for us to determine the calculation model.

We directly grow cobalt hydroxide nanowires (Co(OH)₂) and Co/Mo bimetallic layered double hydroxides (CoMo LDH) nanosheets on carbon cloth.^[8,9] The catalytic activity in artificial alkaline seawater (1 M KOH+0.5 M NaCl) is shown in **Figure R13f**. Compared with CoO, the catalytic activity of Co(OH)₂ and CoMo LDH is obviously poor. The reason for this is mainly attributed to the high intrinsic activity of cobalt hydroxide and CoMo LDH, which triggers both oxygen evolution reaction (OER) and chlorine evolution reaction (CER), resulting in low OER activity. The above results

further corroborate that only the bimetallic hydroxides formed by reconfiguration during the catalytic process can enhance the OER activity.^[10] In addition, our system is not very relevant compared with cobalt hydroxide and other phases with higher intrinsic activity, thus we have not modeled and calculated according to these high intrinsic phases.

The catalysts are prone to initiate the self-restructuring reaction between the catalysts and high-activity nascent intermediates (e.g., O*, HO* and HOO*) under the OER high overpotential, thus forming hydroxyl metal oxides.^[11,12] These reconstructions caused the destroy of catalyst structure, especially the high corrosion of seawater electrolyte, thus eventually most of the catalysts were seriously deactivated.^[13] Interestingly, the surface of the catalyst is reconstructed to form bimetallic layered double hydroxides in the potential range, which is considered as the "real catalyst" of OER.^[14] In addition, a change in the coordination number of Co and Mo was found by analyzing the EXAFS data before and after the reaction of the MoO₃ catalyst (Table S3 in the manuscript). This is mainly attributed to the substitution of some Co ions by Mo, resulting in CoMo LDH, which consists of edge-sharing MO₆ structures (M = Co or Mo). Based on this, we analyzed MoO₃@CoO/CC after the continuous oxygen evolution reaction in detail to determine the CoMo-LDH phase. The formation of CoMo-LDH between the catalyst MoO₃ and CoO interfaces was verified by HRTEM, HAADF, EELS, XPS and XAS (See details analyzed in Fig. 5 of the manuscript).

Although, the performance of the catalyst was improved after the reconstruction. However, unrestricted reconstruction will lead to the deep reconstruction of the catalyst, which will lead to the gradual phase separation of bimetallic layered double hydroxides formed by pre-reconstruction, and then lead to the deactivation of the catalyst. Fig. 5k in the manuscript well proves that MoO₃@CoO/CC catalyst is confined by MoO₃ layer to realize directional reconstruction. On the contrary, CoO catalysts that are not confined by the MoO₃ layer constantly reconstructed, which leads to poor catalyst stability. To this end, the schematic diagram in Supplementary Fig. 22 of the text clearly illustrates this process of change.

Figure R13. Amorpho-molybdenum oxide grown in carbon fiber cloth (MoO_3/CC). (a) XPS test full spectrum analysis. (b) Contents statistics of each element. (c) XPS fine spectrum of Mo 3d. (d) Raman images of MoO_3/CC . (e) XRD diffraction pattern of sample MoO_3/CC after treatment at different temperatures in inert atmosphere. (f) Linear sweep voltammetry (LSV) curves of various samples.

References:

- [1] Mao, S. et al., High-performance bi-functional electrocatalysts of 3D crumpled graphene-cobalt oxide nanohybrids for oxygen reduction and evolution reactions. *Energy Environ. Sci.* **7**, 609-616 (2014).
- [2] Battaglia, C. et al., Hole selective MoO_x contact for silicon solar cells. *Nano Lett.* **14**, 967-971 (2014).
- [3] Liu, H. F. et al., Atomic layer deposition and post-growth thermal annealing of ultrathin MoO_3 layers on silicon substrates: Formation of surface nanostructures. *Appl. Surf. Sci.* **439**, 583-588 (2018).
- [4] Juan, P. C. et al., Plasma-enhanced atomic layer deposition of molybdenum oxides using molybdenum hexacarbonyl as the precursor. *Mater. Chem. Phys.*, **288**, 126395-126405 (2022).
- [5] Xu, H. et al., Ultra-thin MoO_3 film goes wafer-scaled nano-architectonics by atomic layer deposition. *Mater. Des.*, **149**, 135-144 (2018).
- [6] Ramana, C. V. et al., Chemical and electrochemical properties of molybdenum oxide thin films prepared by reactive pulsed-laser assisted deposition. *Chem. Phys. Lett.*, **428**, 114-118 (2006).
- [7] Hashem, A. M. et al., Electrochemical properties of nanofibers α - MoO_3 as cathode materials for Li batteries. *J. Power Sources*, **219**, 126-132 (2012).
- [8] Jiang L. W. et al., Boosting the Stability of Oxygen Vacancies in α - $\text{Co}(\text{OH})_2$ Nanosheets with Coordination Polyhedrons as Rivets for High-Performance Alkaline Hydrogen Evolution Electrocatalyst. *Adv. Energy Mater.* **12**, 2202351 (2022).
- [9] Zhang, M. et al., Spontaneous synthesis of silver nanoparticles on cobalt-molybdenum layer double hydroxide nanocages for improved oxygen evolution reaction. *J. Colloid Interface Sci.*

- 628, 299-307 (2022).
- [10] Guo D, et al., TiN@Co_{5.47}N Composite Material Constructed by Atomic Layer Deposition as Reliable Electrocatalyst for Oxygen Evolution Reaction. *Adv. Funct. Mater.* **31**, 2008511 (2021).
- [11] Kang, J. X. *et al.*, Valence oscillation and dynamic active sites in monolayer NiCo hydroxides for water oxidation. *Nat. Catal.* **4**, 1050-1058 (2021).
- [12] Ye, S. H. *et al.*, Deeply self-reconstructing CoFe(H₃O)(PO₄)₂ to low-crystalline Fe_{0.5}Co_{0.5}OOH with Fe³⁺-O-Fe³⁺ motifs for oxygen evolution reaction. *Appl. Catal. B: Environ.* **304**, 120986 (2022).
- [13] Chen, F.-Y. *et al.*, Stability challenges of electrocatalytic oxygen evolution reaction: From mechanistic understanding to reactor design. *Joule* **5**, 1704-1731 (2021).
- [14] Chala, S. A. *et al.*, Tuning Dynamically Formed Active Phases and Catalytic Mechanisms of In Situ Electrochemically Activated Layered Double Hydroxide for Oxygen Evolution Reaction. *ACS Nano* **15**, 14996-15006 (2021).

Comments #4: Although it is in strong alkali condition, the author uses PEM electrolyzer, which is usually used in pure water or weak acid environment. According to existing articles, anion exchange membrane (AEM) is used in most alkaline electrolyzers. In addition, the author can complete the specific treatment methods of proton exchange membrane for readers.

Response: We are very grateful for your insightful questions. As the reviewer said, AEM electrolytic cell is generally selected for splitting seawater with weak alkalinity in real seawater.^[1] It is worth noting that in the manuscript we mention the use of flow cells using nafion as a separation (See Fig. 7c in the manuscripts for details). In the original text, we chose an amphoteric nafion membrane, which is mainly because the MoO₃@CoO/CC material we designed can effectively adsorb and shield metal cations in seawater, thus improving the service life of the catalyst in the whole electrode (For a detailed description, see the manuscript support information Fig. S31). In other words, the separator in this paper only plays a role in inhibiting the opposite diffusion of gas. Unfortunately, we didn't carefully check this part of the description in the supporting information. The corresponding modification has been added to the manuscript. In order to clarify this problem, we present data from a flow electrolytic cell test using an anion-exchange membrane Tokuyama A201, Pt/C, and MoO₃@CoO/CC as the separator, cathode, and anode, respectively. As shown in **Figure R14**, the curves of

both diaphragm-assembled electrolytic cells tested at an industrial hydrogen production current density of 1.0 A cm^{-2} for 100 h showed good stability. The above results further show that our designed $\text{MoO}_3@\text{CoO}/\text{CC}$ exhibits good performance of direct electrolysis of seawater.

The treatment methods of membrane: The membrane is used directly after being immersed in the corresponding electrolyte for 24 h. The corresponding treatment method has been supplemented in the supporting information.

Figure R14. The stability test of flow electrolytic cell using two kinds of separator assembly.

References:

- [1] Dresp, S. *et al.*, Efficient direct seawater electrolyzers using selective alkaline NiFe-LDH as OER catalyst in asymmetric electrolyte feeds. *Energy Environ. Sci.* **13**, 1725-1729 (2020).

Comments #5: The contents of reconstruction in this manuscript should be discussed together, including HRTEM, in-situ Raman, and reconstruction mechanism diagram, etc.

Response: We thank the reviewer for the insightful question. According to the reviewer's comments, we discuss the reconstruction contents such as HRTEM, in-situ Raman and reconstruction mechanism diagram in the manuscript together. See Fig. 5 of the manuscript for details, and highlighted with a yellow background.

Comments #6: The abscissa difference of CV curve is generally 0.1 V. (Figure 5c, S6,

S7, S13)

Response: We truly appreciate the reviewer's helpful suggestions. We re-tested the CV curves, re-evaluated the electric double layer capacitance of the catalyst, and revised Figure 5c, Figure S6, Figure S7 and Figure S13 (see **Figure R15-R18**), and added them to the manuscript.

Figure R15. The double capacitance (C_{dl}) comparison plots of $\text{MoO}_3@CoO/CC$ catalyst before and after the reaction.

Figure R16. The cyclic voltammograms (CV) of (a) $\text{MoO}_3@CoO/CC$ -100 cy, (b) $\text{MoO}_3@CoO/CC$ -300 cy, (c) $\text{MoO}_3@CoO/CC$ -500 cy, (d) $\text{MoO}_3@CoO/CC$ -600 cy, (e) $\text{MoO}_3@CoO/CC$ -800 cy and (f) $\text{MoO}_3@CoO/CC$ -1000 cy performed in a non-Faradaic regime at different scan rates. (g) Plots

of capacitive currents vs different scan rates with calculated double-layer capacitances C_{dl} .

Figure R17. The cyclic voltammograms (CV) of (a) CoO/CC, (b) MoO₃/CC and (c) MoO₃@CoO/CC performed in a non-Faradaic regime at different scan rates.

Figure R18. The cyclic voltammograms (CV) of (a) MoO₃@CoO/CC-after and (b) CoO/CC-after performed in a non-Faradaic regime at different scan rates. (c) The C_{dl} comparison plots of MoO₃@CoO/CC catalyst before and after the reaction. (d) The C_{dl} comparison plots of CoO/CC catalyst before and after the reaction.

Comments #7: It is obviously unreasonable to infer the mechanism of OER from EPR intensity, and it is suggested to prove the transformation of OER mechanism by other characterization methods, such as the in situ differential electrochemical mass spectrometry (DEMS) and pH-dependent experiment.

Response: Special thanks for your good suggestion to enhance the paper rigor and

persuasiveness. We fully agree with the reviewer that it is obviously unreasonable to infer OER mechanism only from EPR data. In this paper, we first observed the formation of a new phase (CoMo-LDH) between MoO₃ and CoO in MoO₃@CoO/CC after catalyst by TEM (See Fig.5b in the in the manuscript). Subsequently, the structures of the catalyst and the newly formed phase were further confirmed by XPS and XAS (See Fig.5d-h and Fig. S14-S20 in the in the manuscript). Based on the adsorption mechanism (AEM) of OER catalyst, the catalyst will be deeply reconstructed under high potential, thus deactivating the catalyst.^[1,2] However, the catalyst exhibits a long-lasting stability, suggesting that the catalyst does not act according to the AEM mechanism.^[3-6] The main reason for the altered catalyst mechanism is that the MoO₃ layer functions as a directional confined reconfiguration for the catalyst. Then, we used infrared spectroscopy, DFT and in-situ Raman spectroscopy to verify that the layered MoO₃ has the function of directional confined reconstruction of the MoO₃@CoO/CC. The changes of the catalytic mechanism from an AEM to a lattice oxygen mechanism (LOM) in the presence of the confined layer MoO₃ is demonstrated by EPR spectroscopy (See Fig.6e in the in the manuscript) and density of states calculations for the O(2p) orbitals (See Fig.6f in the in the manuscript).

To further clarify this issue, we added tetramethylammonium cation (TMA⁺) detection, differential electrochemical mass spectrometry (DEMS) and pH-dependent experiments according to the reviewer comments. It is very important to detect the O₂²⁻ species produced by LOM during OER by TMA⁺.^[7,8] The OER activity of the MoO₃@CoO/CC catalyst was significantly reduced after the addition of TMAOH to the alkaline electrolyte (**Figure R19a**). This is mainly due to the strong interaction between O₂²⁻ species and TMA⁺, which inhibits the LOM pathway of the catalyst. On the contrary, because CoO/CC catalyst evolution oxygen by AEM mechanism, thus the performance changes are not obvious.^[9] From **Figure R19b**, it is observed that MoO₃@CoO/CC catalyst has two characteristic peaks corresponding to TMA⁺ at 751.7 and 950.6 cm⁻¹, which further proves that its OER process follows LOM mechanism.^[8] In order to clarify the OER process of LOM directly, the MoO₃@CoO/CC catalyst was labeled with ¹⁸O isotope. Firstly, MoO₃@CoO/CC was electrochemically activated

with H_2^{18}O solution in 0.1 M KOH electrolyte. Then, electrolyzed with H_2^{16}O solution after labeling, and the generated oxygen product was verified by DEMS (**Figure R19c**). The activated $\text{MoO}_3@\text{CoO}/\text{CC}$ catalyst has obvious periodic intensity of $^{18}\text{O}^{16}\text{O}$ peak (mass-to-charge ratio, $m/z = 34$), while $^{18}\text{O}^{18}\text{O}$ has no signal ($m/z = 36$). This result suggests that the $\text{MoO}_3@\text{CoO}/\text{CC}$ catalyst mechanism that undergoes activation induced directed reconfiguration transforms into LOM.^[10] In addition, the OER activity of $\text{MoO}_3@\text{CoO}/\text{CC}$ catalyst at different pH values was significantly enhanced with the increase of pH value compared to CoO/CC (**Figure R19d**). The RHE-scaled proton reaction level ($\rho^{\text{RHE}} = \partial \log(j) / \partial \text{pH}$) was used to further elucidate the dependence of the catalyst undergoing an OER reaction on proton activity.^[8] The ρ^{RHE} value of $\text{MoO}_3@\text{CoO}/\text{CC}$ (0.71) is closer to 1 than that of CoO/CC (0.12), which further indicates the pH dependence of OER kinetics (**Figure R19e**), thus proving that the reconstructed $\text{MoO}_3@\text{CoO}/\text{CC}$ follows LOM rather than the traditional AEM mechanism in the OER process.

The supplementary experimental data have been added to the manuscript and highlighted with a yellow background.

Figure R19. (a) LSV curves of MoO₃@CoO/CC and CoO/CC in 1.0 M KOH and 1.0 M TMAOH. (b) Raman spectra of MoO₃@CoO/CC and CoO/CC measured after running at 1.6 V vs RHE for 30 min in 1.0 M TMAOH and 1.0 M KOH solutions and washing with deionized water. (c) The DEMS signals of ³⁴O₂ and ³⁶O₂ for MoO₃@CoO/CC. (d) LSV curves measured in KOH electrolytes with pH = 12.5, 13, 13.5, and 14. (e) At 1.6 V vs RHE plotted in log scale as a function of pH.

References:

- [1] Song, J. *et al.*, A review on fundamentals for designing oxygen evolution electrocatalysts. *Chem. Soc. Rev.* **49**, 2196-2214 (2020).
- [2] Yang, C. *et al.*, Cation insertion to break the activity/stability relationship for highly active oxygen evolution reaction catalyst. *Nat. Commun.* **11**, 1378 (2020).
- [3] Kuai, C. *et al.*, Phase segregation reversibility in mixed-metal hydroxide water oxidation catalysts. *Nat. Catal.* **3**, 743-753 (2020).
- [4] Minguzzi, A., How to improve the lifetime of an electrocatalyst. *Nat. Catal.* **3**, 687-689 (2020).
- [5] Chen, F.-Y., Wu, Z.-Y., Adler, Z. & Wang, H., Stability challenges of electrocatalytic oxygen evolution reaction: From mechanistic understanding to reactor design. *Joule* **5**, 1704-1731 (2021).
- [6] Shi, Z. *et al.*, Confined Ir single sites with triggered lattice oxygen redox: Toward boosted and sustained water oxidation catalysis. *Joule* **5**, 2164-2176 (2021).
- [7] Tan, X. H., *et al.*, Electrochemical Etching Switches Electrocatalytic Oxygen Evolution Pathway of IrO_x/Y₂O₃ from Adsorbate Evolution Mechanism to Lattice-Oxygen-Mediated Mechanism. *Small* **19**, 2303249 (2023).
- [8] Wang, F. Q. *et al.*, Activating lattice oxygen in high-entropy LDH for robust and durable water oxidation. *Nat. Commun.* **14**, 6019 (2023).

- [9] Zhang, N. et al., Lattice oxygen redox chemistry in solid-state electrocatalysts for water oxidation. *Energy Environ. Sci.*, **14**, 4647-4671 (2021).
- [10] Niu, Z. Q. et al., Robust Ru-VO₂ Bifunctional Catalysts for All-pH Overall Water Splitting. *Adv. Mater.*, **36**, 2023, 2310690 (2023).

Comments #8: Please explain "Compared with Cl⁻, OH⁻ with smaller radius easily passes through MoO₃ layer." and "The CoMo LDH layered ...which has strong electrostatic repulsion to Cl⁻" in detail. Does the anti-chlorine corrosion effect of MoO₃ layer come from its passivation film or electrostatic attraction? In addition, does MoO₃ lead to the formation of MoO₄²⁻ under alkaline conditions? (Lines 222 and 281)

Response: Thanks for the reviewer's comments. The excellent catalytic activity and stability of MoO₃@CoO/CC catalyst can be attributed to the fact that the customized MoO₃ layer can shield Cl⁻ and adjust the interface, thus realizing selective seawater oxidation. The key to the chlorine corrosion resistance of the MoO₃@CoO/CC catalyst lies in the selectivity of MoO₃ layer through OH⁻. The MoO₃ layer can achieve selectivity mainly due to the following points: 1) Compared with Cl⁻, OH⁻ with smaller radius easily passes through MoO₃ layer.^[1] 2) It is verified by theoretical calculations that the migration energy barrier of Cl⁻ in the MoO₃ layer is much higher than that of H₂O/OH⁻ (See Fig. 4c-f in the manuscript for detailed analysis). 3) For heterogeneous catalysis, the first step of the reaction is adsorption,^[2] and the calculation shows that the adsorption energy of MoO₃ for OH⁻ is much higher than that of Cl⁻. In addition, the formation energy of the MoO₃ layer with Cl⁻ is small, indicating that the MoO₃ layer is inert to Cl⁻.^[3] In the process of seawater oxidation, the catalyst is directionally reconstructed between MoO₃ and CoO to form CoMo LDH (See Fig. 5 in the manuscript). CoMo LDH is a crystal structure with the metal in the centre of the octahedron six oxygen atoms on the outside,^[4,5] thus providing electrostatic repulsion to Cl⁻. For this reason, the formed CoMo-LDH and the MoO₃ layer synergistically further increase the Cl⁻ shielding ability.

In the Mo 3dXPS spectrum (**Figure R20**), a doublet is observed at 232.7 and 235.8 eV (Mo⁶⁺ 3d_{5/2} and Mo⁶⁺ 3d_{3/2}), which originates from MoO₃@CoO/CC. Because Mo⁶⁺ in MoO₃ is in the highest valence state, it will not lead to the formation of MoO₄²⁻ under

alkaline conditions.^[6]

Figure R20. The high-resolution XPS spectra of Mo 3d for the MoO₃@CoO/CC catalysts.

References:

- [1] Huang, C., et al., The debut and spreading the landscape for excellent vacancies-promoted electrochemical energy storage of nano-architected molybdenum oxides. *Mater. Today Energy* **30**, 101154 (2022).
- [2] Yang, C. *et al.*, Cation insertion to break the activity/stability relationship for highly active oxygen evolution reaction catalyst. *Nat. Commun.* **11**, 1378 (2020).
- [3] Xu, X. W. et al., Corrosion-resistant cobalt phosphide electrocatalysts for salinity tolerance hydrogen evolution. *Nat. Commun.* **14**, 7708 (2023).
- [4] Shen, W. *et al.*, Defect engineering of layered double hydroxide nanosheets as inorganic photosensitizers for NIR-III photodynamic cancer therapy. *Nat. Commun.* **13**, 3384 (2022).
- [5] Bao, J. et al. The CoMo-LDH ultrathin nanosheet as a highly active and bifunctional electrocatalyst for overall water splitting. *Inorg. Chem. Front.* **5**, 2964–2970 (2018).
- [6] Kang, X. et al., A corrosion-resistant RuMoNi catalyst for efficient and long-lasting seawater oxidation and anion exchange membrane electrolyzer. *Nat. Commun.* **14**, 3607 (2023).

Reviewer 3: The author synthesized ultra-thin amorphous MoO₃ using ALD technology on the surface of beaded CoO. At the MoO₃ and CoO interface, a new phase, identified as CoMo LDH after OER, demonstrated remarkable efficacy in hindering extensive interphase reconstruction, thereby enhancing the protection of genuine active sites. However, the current characterization and computational analysis mechanisms lack adequacy in substantiating this concept, making it challenging to endorse in

esteemed journals like Nature Communications. The author should carefully address the following suggestions:

Author's response: We thank the reviewer's high evaluation and valuable suggestions of our work. We have carried out supplementary experiments and have made modifications according to the reviewer's constructive and valuable suggestions.

Comments #1: The identification of amorphous MoO₃ remains unclear. Firstly, the author did not synthesize crystalline MoO₃ using similar methods. Secondly, there is a lack of related characterization to systematically identify MoO₃.

Response: We appreciate the reviewer's comments on the MoO₃ phase and have added relevant description of our experiments in details, as well as performed additional tests to address the reviewer's concern. The amorphous MoO₃ structure constructed by ALD mainly determines the through the following experiments: From the XPS spectrum of MoO_x thin films prepared by ALD on carbon cloth, it can be seen that they mainly contain Mo, C and O elements (**Figure R21a**). The contents of various elements analyzed by XPS are shown in **Figure R21b**. The results show that the atomic contents of Mo and O in the prepared MoO_x are close to 1:3, indicating that the phase of the prepared material is MoO₃. In **Figure R21c**, the XPS spectrum shows double peaks at 232.7 and 235.8 eV, corresponding to Mo 3d^{5/2} and Mo 3d^{3/2},^[1,2] respectively. These two peaks belong to the completely oxidized Mo⁶⁺ state, corresponding to the stoichiometric structure of MoO₃. It can be seen that the wavelength at 302, 433, 619, 670, 824, 963 cm⁻¹ corresponds to the Mo-O bond in MoO_x from the Raman spectrum (**Figure R21d**), which indicates the presence of MoO₃ species.^[3,4] From **Figure R21e** that different annealing temperatures have great influence on the crystallization of the film. When the annealing temperature reaches 400 °C, α-MoO₃ diffraction peaks appear at 12.8°, 25.7° and 39.1°, and β-MoO₃ diffraction peaks appear at 23° and 25°.^[5,6] With the temperature rising to 500 °C, the XRD peak of β-MoO₃ disappeared, indicating that the purity of the film is higher.^[6] The above analysis verified that amorphous MoO₃ was prepared by ALD.

The Mo-K edge analysis in XAS in the manuscript shows that the molybdenum

oxide prepared by ALD has the same valence state compared to pure MoO₃ (See Fig. 5h in the manuscript), further verifying that ALD synthesized sample is MoO₃ phase. Besides, the coordination structure of Mo atoms in the as-prepared samples was further analyzed by EXAFS. Based on EXAFS quantitative fitting (Table S3 in the manuscript), the first shell of central atoms Mo shows coordination numbers is 6. The above analysis shows MoO₃ prepared by ALD is an octahedral structure of MoO₆ based on EXAFS data.

Figure R21. Amorpho-molybdenum oxide grown in carbon cloth (MoO₃/CC). (a) XPS test full spectrum analysis. (b) Contents statistics of each element. (c) XPS fine spectrum of Mo 3d. (d) Raman images of MoO₃/CC. (e) XRD diffraction pattern of sample MoO₃/CC after treatment at different temperatures in inert atmosphere.

References:

- [1] Battaglia, C. et al., Hole selective MoO_x contact for silicon solar cells. *Nano Lett.* **14**, 967-971 (2014).
- [2] Liu, H. F. et al., Atomic layer deposition and post-growth thermal annealing of ultrathin MoO₃ layers on silicon substrates: Formation of surface nanostructures. *Appl. Surf. Sci.* **439**, 583-588 (2018).
- [3] Juan, P. C. et al., Plasma-enhanced atomic layer deposition of molybdenum oxides using molybdenum hexacarbonyl as the precursor. *Mater. Chem. Phys.*, **288**, 126395-126405 (2022).
- [4] Xu, H. et al., Ultra-thin MoO₃ film goes wafer-scaled nano-architectonics by atomic layer deposition. *Mater. Des.*, **149**, 135-144 (2018).
- [5] Ramana, C. V. et al., Chemical and electrochemical properties of molybdenum oxide thin films prepared by reactive pulsed-laser assisted deposition. *Chem. Phys. Lett.*, **428**, 114-118 (2006).

[6] Hashem, A. M. et al., Electrochemical properties of nanofibers α -MoO₃ as cathode materials for Li batteries. *J. Power Sources*, **219**, 126-132 (2012).

Comments #2: In Fig. 2e, the author associates peaks at 531.48 and 532.51 eV with surface chemically adsorbed water for MoO₃@CoO/CC and MoO₃/CC. However, a small peak near 532.51 eV is observed for both MoO₃@CoO/CC and CoO/CC. Confirm the accuracy of the XPS analysis.

Response: We thank the reviewer for the insightful questions and valuable suggestions. As suggested by the reviewers, we re-tested the corresponding samples. In **Figure R22**, there are three obvious peaks at 529.9, 531.4 and 532.4 eV, which are attributed to the binding energy of metal oxygen bond (M-O), -OH and surface chemically adsorbed water (O_{ads}),^[1,2] respectively. Compared with MoO₃/CC and CoO/CC, the two characteristic peaks of MoO₃@CoO/CC have obvious negative shifts, indicating strong electronic interactions between MoO₃ and CoO. Furthermore, the peak of MoO₃@CoO/CC is stronger at 531.4 eV, which indicates that ALD MoO₃ optimizes the adsorption energy of CoO to H₂O.

The corresponding data are supplemented in the manuscript Figure 2e.

Figure R22. The high-resolution XPS spectra of O 1s from various samples.

Table R3. Detailed fitting information of O 1s in various samples.

Samples	Detailed fitting information of O 1s in MoO ₃ @CoO/CC samples		
Peak	529.8 eV	531.5 eV	532.3
Area	23311.32	13910.78	10005.86
FWHM	1.18	1.21	1.87
Samples	Detailed fitting information of O 1s in CoO/CC samples		
Peak	530.7 eV	531.6	532.6 eV
Area	48210.62	14157.62	8330.1
FWHM	1.1	1.0	1.8
Samples	Detailed fitting information of O 1s in MoO ₃ /CC samples		
Peak	530.5 eV	531.6	532.6 eV
Area	40435.97	8243.27	8249.287
FWHM	1.43	1.21	1.64

References:

- [1] Wu, L. *et al.*, Boron-modified cobalt iron layered double hydroxides for high efficiency seawater oxidation. *Nano Energy* **83**, 105838 (2021).
- [2] Yang, G. *et al.*, Interfacial Engineering of MoO₂-FeP Heterojunction for Highly Efficient Hydrogen Evolution Coupled with Biomass Electrooxidation. *Adv. Mater.* **32**, 2000455 (2020).

Comments #3: In the DFT section, unify the notation for MoO₃@CoO/CC and MoO₃@CoO.

Response: Special thanks for your good suggestion to enhance the paper rigor and persuasiveness. These mistakes in the paper have been corrected and highlighted with a yellow background.

Comments #4: Figure 4 explores the migration energy barrier of H₂O/OH and Cl⁻ for MoO₃@CoO/CC and CoO/CC. It is crucial to supplement the associated computation about pure MoO₃, as the ionic migration path for MoO₃@CoO/CC is within MoO₃ in this article.

Response: We thank the reviewer for the insightful questions and valuable suggestions. According to the reviewer's suggestion, we calculated the migration energy barrier of H₂O/OH and Cl⁻ in pure MoO₃. As shown in **Figure R23**, a structural model of the

migration path of $\text{Cl}^-/\text{H}_2\text{O}/\text{OH}^-$ in various samples is constructed. As the ultra-thin MoO_3 layer is deposited on the surface of CoO , the adsorption energy of the catalyst to reactants/intermediates is effectively regulated, thereby the migration energy barrier of $\text{H}_2\text{O}/\text{OH}^-$ at the $\text{MoO}_3@/\text{CoO}/\text{CC}$ interface lower. As assumed, the migration energy barrier of $\text{H}_2\text{O}/\text{OH}^-$ at $\text{MoO}_3@/\text{CoO}/\text{CC}$ interface is lower than that at CoO/CC and MoO_3/CC interface (**Figure R23b, c**). On the contrary, compared with CoO interface, the migration energy barrier of Cl^- at $\text{MoO}_3@/\text{CoO}/\text{CC}$ and MoO_3/CC interface is higher (**Figure R23d**). These results verify that MoO_3 layer exhibits the ability to shield chloride ions, which provides the possibility for selective catalysis.

The corresponding modifications have been added to the manuscript.

Figure R23. (a) The migration paths of Cl^- , H_2O and OH^- in various structure. The migration energy barrier of (b) H_2O , (c) OH^- and (d) Cl^- in various structure.

Comments #5: Although the article describes MoO_3 as amorphous, the DFT model depicts periodic MoO_3 in $\text{MoO}_3@/\text{CoO}/\text{CC}$. Explain the correction of the model for the DFT calculation.

Response: We are appreciated of your kind advice that encourages us to improve the quality of this manuscript. Amorphous structure is a kind of short-range order or long-range disorder, which forms a structure similar to atomic cluster structure in small size and relatively random and disorderly accumulation of atomic clusters in large size.^[1,2]

In fact, we have constructed amorphous MoO_3 in the computational modeling of density functional theory (DFT), but because the model in the manuscript gives a side structure that is easy to be regarded as ordered. To clarify this problem, we give structural models from different perspectives as shown in **Figure R24**. For this, the corresponding calculation model in this paper has adjusted the angle.

Figure R24. Density functional theory (DFT) calculation model. The structure of catalyst was observed from different angles.

References:

- [1] Wan, G. *et al.*, Amorphization mechanism of SrIrO_3 electrocatalyst: How oxygen redox initiates ionic diffusion and structural reorganization. *Sci. Adv.* **7**, eabc7323 (2021).
- [2] Liu, S. H. *et al.*, A top-down strategy for amorphization of hydroxyl compounds for electrocatalytic oxygen evolution. *Nat. Commun.* **13**, 1187 (2022).

Comments #6: Corroborate the accuracy of the experimental model by fitting the DFT model with EXAFS data (Small 2021, 17, 2101163, DOI: 10.1002/sml.202101163).

Response: We thank the reviewer for the insightful questions and valuable suggestions. Fig. 5f shows the Co-k edge X-ray absorption near edge structures (XANES) for Co foil, $\text{MoO}_3@CoO/CC$, $\text{MoO}_3@CoO/CC$ -after and CoO. The absorption edge of $\text{MoO}_3@CoO/CC$ is located near CoO (inset in Fig. 5f), indicating that the Co valence state in $\text{MoO}_3@CoO/CC$ is close to that of CoO. This is mainly due to the formation of chemical bonds between MoO_3 and CoO, which makes the Co valence state in $\text{MoO}_3@CoO/CC$ higher than that of pure CoO. This further verified that MoO_3 was anchored on the CoO surface by Co-O-Mo bond. Note that since the key signals of Co-O-Co and Co-O-Mo are close, they are collectively referred to as Co-O-Co/Mo. The

coordination peaks of Co-O and Co-O-Co in the CoO reference samples were found to be 1.38 and 2.67 Å in the Fourier transform (FT) k^3 -weighted extended X-ray absorptiometry (EXAFS) spectrum (Fig. 5g). The EXAFS of MoO₃@CoO/CC and CoO/CC are quantitatively fitted to obtain specific structural parameters, and the fitting results are shown in **Table R4**. Based on EXAFS quantitative fitting, the first shell of central atoms Co shows coordination numbers is 6.^[1,2] As shown in **Figure R25a**, Co is a cubic crystal with (Fm-3m(225)) spatial lattice by EXAFS data.

Fig. 5h shows the Mo-k edge XANES for MoO₃@CoO/CC, MoO₃@CoO/CC-after, Mo foil, MoCl₅ and MoO₃. Among them, the K-edge of MoO₃@CoO/CC is between Mo foil and MoO₃, indicating that the valence of Mo at between them. Fig. 5i shows that the peak at 1.5 Å corresponds to Mo-O bond in MoO₃. The Mo-O bond in MoO₃@CoO/CC shifts negatively by 0.15 Å compared to MoO₃. The reason for this phenomenon is mainly attributed to the existence of partial Co-O-Mo bonds in MoO₃@CoO/CC. Notably, due to the bond signals of Mo-O-Co and Mo-O are close, they are collectively referred to as Mo-O-Co/Mo-O. The coordination structure of Mo atoms in the as-prepared samples was further analyzed by EXAFS. Based on EXAFS quantitative fitting (**Table R4**), the first shell of central atoms Mo shows coordination numbers is 6. As shown in **Figure R25b**, MoO₃ prepared by ALD is an octahedral structure of MoO₆^[3,4] based on EXAFS data.

The above analysis is consistent with DFT calculation model.

Figure R25. Corroborate the accuracy of the experimental model by fitting the DFT model with EXAFS data.

Table R4. Structural parameters extracted from the EXAFS fitting. EXAFS fitting parameters at the M K-edge (M=Co, Mo) for various samples.

Sample	Shell	CN ^a	R(Å) ^b	σ ² (Å ²) ^c	ΔE ₀ (eV) ^d	R factor
Co K-edge (S ₀ ² =0.753)						
Co foil	Co-Co	12*	2.494±0.003	0.0061±0.0001	6.7±0.3	0.0029
CoO	Co-O	6.0±0.4	2.077±0.022	0.0102±0.0018	-2.1±0.9	0.0063
	Co-Co	11.8±0.5	3.016±0.005	0.0081±0.0008		
MoO ₃ @CoO/CC	Co-O1	0.8±0.5	1.985±0.013	0.0082±0.0024	1.5±3.6	0.0081
	Co-O2	4.7±0.3	2.105±0.026			
	Co-Co	10.1±0.8	3.010±0.017	0.0094±0.0005	1.2±2.1	
	Co-O-Mo	1.2±0.3	3.198±0.024			
MoO ₃ @CoO/CC-after	Co-O1	2.4±0.3	1.906±0.017	0.0052±0.0045	2.1±6.1	0.0137
	Co-O2	3.4±0.1	2.134±0.016			
	Co-Co	9.7±0.5	3.003±0.019	0.0105±0.0021	1.6±6.9	
	Co-O-Mo	0.8±0.3	3.213±0.016			
Mo K-edge (S ₀ ² =0.803)						
Mo foil	Mo-Mo	8*	2.721±0.005	0.0031±0.0008	-8.4±0.7	0.0005
	Mo-Mo	6*	3.124±0.003	0.0024±0.0008		
MoO ₃	Mo-O	6.0±0.5	1.956±0.005	0.0034±0.0005	-2.4±1.0	0.0078
	Mo-Mo	5.0±0.7	3.651±0.004	0.0032±0.0004		
MoO ₃ @CoO/CC	Mo-O	4.8±0.5	1.846±0.011	0.0106±0.0046	-3.4±6.8	0.0086
	Mo-Mo	5.0±0.2	3.147±0.005	0.0023±0.0006		
	Mo-O-Co	2.3±0.3	1.78±0.011	2.6±0.6	-6.7±0.6	
MoO ₃ @CoO/CC-after	Mo-O	4.5±0.2	2.03±0.005	0.0031±0.0014	2.9±1.4	0.0043
	Mo-Mo	4.2±0.2	2.52±0.002	0.0056±0.0003		
	Mo-O-Co	2.4±0.1	1.04±0.08	3.17±0.046	-3.1±0.6	
	Mo-Cl	2.8±0.1	1.76±0.09	2.25±0.038		
MoCl ₅	Mo-Mo	5.2±0.8	2.12±0.008	0.051±0.0007	-5.2±1.5	0.0025
	Mo-Cl	5.1±0.2	2.14±0.05	4.14±0.06		

^aCN, coordination number; ^bR, the distance to the neighboring atom; ^cσ², the Mean Square Relative Displacement (MSRD); ^dΔE₀, inner potential correction; R factor indicates the goodness of the fit. S₀² was fixed to 0.753 and 0.803, according to the experimental EXAFS fit of Co foil and Mo foil by fixing CN as the known crystallographic value. * This value was fixed during EXAFS fitting, based on the known structure of Co and Mo. Fitting range: 3.0 ≤ k (Å⁻¹) ≤ 14.0 and 1.0 ≤ R (Å) ≤ 3.0 (Co foil); 2.0 ≤ k (Å⁻¹) ≤ 13.6 and 1.0 ≤ R (Å) ≤ 3.5 (MoO₃@CoO/CC); 2.0 ≤ k (Å⁻¹) ≤ 13.5 and 1.0 ≤ R (Å) ≤ 3.5 (MoO₃@CoO/CC-after); 3.0 ≤ k (Å⁻¹) ≤ 12.0 and 1.0 ≤ R (Å) ≤ 3.0 (Mo foil); 2.0 ≤ k

(\AA) ≤ 12.7 and $1.0 \leq R$ (\AA) ≤ 3.0 (MoO_3); $2.0 \leq k$ ($/\text{\AA}$) ≤ 12.7 and $1.0 \leq R$ (\AA) ≤ 3.0 (MoCl_5). A reasonable range of EXAFS fitting parameters: $0.700 < S_0^2 < 1.000$; $CN > 0$; $\sigma^2 > 0 \text{\AA}^2$; $|\Delta E_0| < 10 \text{ eV}$; R factor < 0.02 .

References:

- [1] Xue, H., Meng, A., Yang, T., Li, Z. & Chen, C., Controllable oxygen vacancies and morphology engineering: Ultra-high HER/OER activity under base–acid conditions and outstanding antibacterial properties. *J. Energy Chem.* 71, 639–651 (2022).
- [2] Tian, Y. *et al.*, Engineering Crystallinity and Oxygen Vacancies of Co(II) Oxide Nanosheets for High Performance and Robust Rechargeable Zn–Air Batteries. *Adv. Funct. Mater.* 31, 2101239 (2021).
- [3] Shen, W. *et al.*, Defect engineering of layered double hydroxide nanosheets as inorganic photosensitizers for NIR-III photodynamic cancer therapy. *Nat. Commun.* 13, 3384 (2022).
- [4] Bao, J. *et al.* The CoMo-LDH ultrathin nanosheet as a highly active and bifunctional electrocatalyst for overall water splitting. *Inorg. Chem. Front.* 5, 2964–2970 (2018).

Comments #7: In Fig. 5b, the author identifies darker areas as CoMo LDH based on lattice spacing. This characterization is insufficient; additional techniques such as EELS should be implemented.

Response: Special thanks for your good suggestion to enhance the paper rigor and persuasiveness. In this paper, we first observed the formation of a phase (CoMo -LDH) between MoO_3 and CoO in $\text{MoO}_3@\text{CoO}/\text{CC}$ -after catalyst by HRTEM (See Fig.5b in the in the manuscript). Subsequently, the structures of the catalyst and the newly formed phase were further confirmed by EELS, XPS and XAS (See Fig.5c-i and Fig. S13-S21 in the in the manuscript). In addition, a change in the coordination number of Co and Mo was found by analyzing the EXAFS data before and after the reaction of the MoO_3 catalyst (Table S3 in the manuscript). This is mainly attributed to the substitution of some Co ions by Mo, resulting in CoMo LDH, which consists of edge-sharing MO_6 structures ($M = \text{Co}$ or Mo).^[1-3]

To further clarify this issue, we supplemented the EELS spectra of the $\text{MoO}_3@\text{CoO}/\text{CC}$ -after catalysts as shown in **Figure R26**. As shown in **Figure R26a**, the green and dark yellow wireframes are the surface scan and error margin,

respectively. As shown in **Figure 26Rb**, the outermost layer, the interfacial and intermediate layer of the MoO₃@CoO-after sample was selected, correspond to the MoO₃, CoMo-LDH and CoO, respectively. In **Figure 26Rc**, the peak at 535.5 eV detected in the MoO₃ layer corresponds to the Mo-O bond in the amorphous structure.^[4] The shoulder peak at 531 eV is attributed to the hybridisation of O 2p with Co 3d and Mo 3d orbitals.^[4] It is worth noting that the hybridization intensities between O 2p and M 3d orbitals is weak, which is mainly due to the influence of -O-O- structure.^[4,5] The Co-L3 edge of the interfacial layer is blue-shifted by about 0.6 eV compared to the intermediate-phase CoO, indicating a higher Co valence.^[5] Similarly, the Mo edge of the interface layer is blue shifted by about 2.0 eV compared to the MoO₃ layer, indicating that the MoO₃ is partially reconstructed.^[6,7] The above results further verify the existence of CoMo-LDH.

Figure R26. (a) HAADF image of MoO₃@CoO after continuous oxygen evolution in seawater for 50 h. (b) EELS Spectrum Image (low-loss). EELS spectra of (c) O-K, (d) Co L-edge and (e) Mo M-edge of MoO₃@CoO/CC-after.

References:

- [1] Zhang, M. et al., Spontaneous synthesis of silver nanoparticles on cobalt-molybdenum layer double hydroxide nanocages for improved oxygen evolution reaction. *J. Colloid Interface Sci.* **628**, 299-307 (2022).
- [2] Shen, W. et al., Defect engineering of layered double hydroxide nanosheets as inorganic photosensitizers for NIR-III photodynamic cancer therapy. *Nat. Commun.* **13**, 3384 (2022).
- [3] Bao, J. et al. The CoMo-LDH ultrathin nanosheet as a highly active and bifunctional electrocatalyst for overall water splitting. *Inorg. Chem. Front.* **5**, 2964–2970 (2018).
- [4] Hu, Y. et al., Understanding the sulphur-oxygen exchange process of metal sulphides prior to oxygen evolution reaction. *Nat. Commun.* **14**, 1949 (2023).
- [5] Chen, J. S. et al., Co–Fe–Cr (oxy)Hydroxides as Efficient Oxygen Evolution Reaction Catalysts. *Adv. Energy Mater.* **11**, 2003412 (2021).

- [6] Zhang, L. L. et al., Tuning Electrical Conductance in Bilayer MoS₂ through Defect-Mediated Interlayer Chemical Bonding. *ACS Nano* **14**, 10265–10275 (2020).
- [7] Esmacilrad, M. et al., Imidazolium-functionalized Mo₃P nanoparticles with an ionomer coating for electrocatalytic reduction of CO₂ to propane. *Nat. Energy* **8**, 891–900 (2023).

Comments #8: Explain how the author identified CoMo LDH as an octahedral structure.

Response: Thanks for the reviewer's comments. In the process of seawater oxidation, the catalyst is directionally reconstructed between MoO₃ and CoO to form CoMo LDH (See Fig. 5 in the manuscript). Importantly, a lattice spacing of 0.38 nm was observed between the MoO₃ layer and CoO (Fig. 5b1), corresponding to the (006) crystal plane of CoMo LDH.^[1,2] This result is consistent with the (006) phase of CoMo-LDH (JCPDS no. 46-0605). In addition, the phase of CoMn-LDH were also confirmed through the EELS, XPS and XAS (See Fig. 5c-i in the manuscript). CoMo LDH is an octahedral structure with metal (M=Co, Mo) in the centre of the octahedron and six oxygen atoms on the outside, which was composed of the edgesharing MO₆ structure (M = Co or Mo).^[1,2]

References:

- [1] Shen, W. *et al.*, Defect engineering of layered double hydroxide nanosheets as inorganic photosensitizers for NIR-III photodynamic cancer therapy. *Nat. Commun.* **13**, 3384 (2022).
- [2] Bao, J. et al. The CoMo-LDH ultrathin nanosheet as a highly active and bifunctional electrocatalyst for overall water splitting. *Inorg. Chem. Front.* **5**, 2964–2970 (2018).

Comments #9: In Fig. 6, the article discusses the adsorbate evolution mechanism (AEM) and lattice-oxygen-mediated mechanism (LOM). Experimental verification is lacking; provide more experimental information following the reference (*Adv. Mater.* **2023**, 2310690, DOI: 10.1002/adma.202310690).

Response: Special thanks for your good suggestion to enhance the paper rigor and persuasiveness. To further clarify this issue, we added tetramethylammonium cation (TMA⁺) detection, differential electrochemical mass spectrometry (DEMS) and pH-dependent experiments according to the reviewer comments. It is very important to detect the O₂²⁻ species produced by LOM during OER by TMA⁺.^[1,2] The OER activity

of the MoO₃@CoO/CC catalyst was significantly reduced after the addition of TMAOH to the alkaline electrolyte (**Figure R27a**). This is mainly due to the strong interaction between O₂²⁻ species and TMA⁺, which inhibits the LOM pathway of the catalyst. On the contrary, because CoO/CC catalyst evolution oxygen by AEM mechanism, thus the performance changes are not obvious.^[3] From **Figure R27b**, it is observed that MoO₃@CoO/CC catalyst has two characteristic peaks corresponding to TMA⁺ at 751.7 and 950.6 cm⁻¹, which further proves that its OER process follows LOM mechanism.^[2] In order to clarify the OER process of LOM directly, the MoO₃@CoO/CC catalyst was labeled with ¹⁸O isotope. Firstly, MoO₃@CoO/CC was electrochemically activated with H₂¹⁸O solution in 0.1 M KOH electrolyte. Then, electrolyzed with H₂¹⁶O solution after labeling, and the generated oxygen product was verified by DEMS (**Figure R27c**). The activated MoO₃@CoO/CC catalyst has obvious periodic intensity of ¹⁸O¹⁶O peak (mass-to-charge ratio, m/z = 34), while ¹⁸O¹⁸O has no signal (m/z = 36). This result suggests that the MoO₃@CoO/CC catalyst mechanism that undergoes activation induced directed reconfiguration transforms into LOM.^[4] In addition, the OER activity of MoO₃@CoO/CC catalyst at different pH values was significantly enhanced with the increase of pH value compared to CoO/CC (**Figure R27d**). The RHE-scaled proton reaction level ($\rho^{\text{RHE}} = \partial \log(j) / \partial \text{pH}$) was used to further elucidate the dependence of the catalyst undergoing an OER reaction on proton activity.^[2] The ρ^{RHE} value of MoO₃@CoO/CC (0.71) is closer to 1 than that of CoO/CC (0.12), which further indicates the pH dependence of OER kinetics (**Figure R27e**), thus proving that the reconstructed MoO₃@CoO/CC follows LOM rather than the traditional AEM mechanism in the OER process.

The supplementary experimental data have been added to the manuscript and highlighted with a yellow background.

Figure R27. (a) LSV curves of MoO₃@CoO/CC and CoO/CC in 1.0 M KOH and 1.0 M TMAOH. (b) Raman spectra of MoO₃@CoO/CC and CoO/CC measured after running at 1.6 V vs RHE for 30 min in 1.0 M TMAOH and 1.0 M KOH solutions and washing with deionized water. (c) The DEMS signals of ³⁴O₂ and ³⁶O₂ for MoO₃@CoO/CC. (d) LSV curves measured in KOH electrolytes with pH = 12.5, 13, 13.5, and 14. (e) At 1.6 V vs RHE plotted in log scale as a function of pH.

References:

- [1] Tan, X. H., et al., Electrochemical Etching Switches Electrocatalytic Oxygen Evolution Pathway of IrO_x/Y₂O₃ from Adsorbate Evolution Mechanism to Lattice-Oxygen-Mediated Mechanism. *Small* **19**, 2303249 (2023).
- [2] Wang, F. Q. et al., Activating lattice oxygen in high-entropy LDH for robust and durable water oxidation. *Nat. Commun.* **14**, 6019 (2023).
- [3] Zhang, N. et al., Lattice oxygen redox chemistry in solid-state electrocatalysts for water oxidation. *Energy Environ. Sci.*, **14**, 4647-4671 (2021).
- [4] Niu, Z. Q. et al., Robust Ru-VO₂ Bifunctional Catalysts for All-pH Overall Water Splitting. *Adv. Mater.*, **36**, 2023, 2310690 (2023).

REVIEWERS' COMMENTS

Reviewer #1 (Remarks to the Author):

This manuscript designed a cowpea-shaped electrocatalyst (MoO₃@CoO//CC) for direct (accurately alkaline but not natural) seawater oxidation. After revision, I do not think this work is greatly improved. So, I would like not to recommend its publication again. The reason is as follow. There are quite a few publications about alkaline seawater splitting. Compared with those published in recent years, there is no enough progress for the article worth publishing in Nature Communications. Firstly, the materials MoO₃ and CoO have been investigated for OER (Appl. Catal. B-Environ. 2023, 328: 122488; etc.). Secondly, the concept about structure reconstruction for improving OER electrocatalysts has been mentioned before (Small DOI: 10.1002/smll.202308613; Chem. Eng. J. 2022, 430: 132632; etc.). Thirdly, the performance, especially the stability, of the AEM cell including the electrocatalyst as anode and Pt/C as cathode is not competitive with those published (Adv. Energy Mater. 2023, 2301475; etc). Therefore, this work is not novel for publication in this top journal, which is more suitable for a professional journal.

Reviewer #2 (Remarks to the Author):

The authors have addressed all the comments and the revised version is recommended for publication.

Reviewer #3 (Remarks to the Author):

The authors have addressed all my concerns, and the paper is ready for publication

Reviewer #1 (Remarks to the Author):

This manuscript designed a cowpea-shaped electrocatalyst ($\text{MoO}_3\text{@CoO/CC}$) for direct (accurately alkaline but not natural) seawater oxidation. After revision, I do not think this work is greatly improved. So, I would like not to recommend its publication again. The reason is as follow. There are quite a few publications about alkaline seawater splitting. Compared with those published in recent years, there is no enough progress for the article worth publishing in Nature Communications. Firstly, the materials MoO_3 and CoO have been investigated for OER (Appl. Catal. B-Environ. 2023, 328: 122488; etc.). Secondly, the concept about structure reconstruction for improving OER electrocatalysts has been mentioned before (Small DOI: 10.1002/sml.202308613; Chem. Eng. J. 2022, 430: 132632; etc.). Thirdly, the performance, especially the stability, of the AEM cell including the electrocatalyst as anode and Pt/C as cathode is not competitive with those published (Adv. Energy Mater. 2023, 2301475; etc.). Therefore, this work is not novel for publication in this top journal, which is more suitable for a professional journal.

Response: Thank you very much for your comments. In this paper, we designed a cowpea-shaped electrocatalyst ($\text{MoO}_3\text{@CoO/CC}$) for direct seawater oxidation. To be precise, the performance in alkaline seawater is studied first, and then the natural seawater electrolysis is directly realized (See Fig. 3~Fig. 7 of the manuscript for details). This is mainly due to: The pH value of real seawater is around 8.0, which is a weak alkaline environment.^[1,2,3] At present, researchers mostly use alkaline artificial seawater to simulate real seawater.^[4-9] Besides, the concentration of salt will gradually increase with the continuous electrolysis of seawater in practical electrolysis applications.^[10] For this reason, the research concept of this paper is to firstly investigate the electrical properties of $\text{MoO}_3\text{@CoO/CC}$ catalysts under artificial alkaline seawater, and then to further investigate the applicability of the catalysts by increasing the alkaline and salt concentrations as well as under real seawater. In the

text, the real seawater comes from Dongtou District, Wenzhou, Zhejiang, China, and is directly used after filtering to remove sediment (See supporting information line 167 for details). After revision, we further clarified the phase and thickness of ALD MoO_3 and the role of blocking chloride ions to achieve selective catalysis (See Fig. 4 of the manuscript for details). Moreover, it is further verified that MoO_3 can inhibit the infinite reconfiguration of the catalyst and realize the dynamic self-reconfiguration of the interface restricted (See Fig. 5 of the manuscript for details). The data support for the transition from the AEM catalyst mechanism to the LOM mechanism has been improved (See Fig. 6 of the manuscript for details). For this reason, the article has been greatly improved after revision.

Although some MoO_3 and CoO materials have been reported for OER (Appl. Catal. B-Environ. 2023, 328: 122488; etc.). However, there are innovations in our paper, as follows: (1) We designed a cowpea-shaped electrocatalyst ($\text{MoO}_3@ \text{CoO}/\text{CC}$). MoO_3 was anchored on the surface of CoO by Mo-O-Co bond, forming a material with a core-shell structure. This is different from the composite of MoO_3 and CoO_x or molybdenum mixed with CoO_x in the literature.^[11, 12] (2) The amorphous MoO_3 phase is designed and constructed by ALD. Meanwhile, CoO_x is CoO phase. (3) In this paper, amorphous MoO_3 layer is mainly used as a chloride ion barrier, which can selectively pass through the reaction substances. Meanwhile, the amorphous MoO_3 layer regulate the interface of CoO and improve the catalytic activity. The active site of water oxidation reaction is CoO interface modified by MoO_3 . In addition, the molybdenum oxide layer can confine the depth reconstruction of cobalt oxide interface, thus improving the catalytic stability. These are not reported in other studies.

During the OER process, the catalyst surface will be oxidized at high anode potential and self-reconstructed to form hydrogen (hydroxyl) oxide.^[13-15] Interestingly, the surface of the catalyst is reconstructed to form hydroxyl metal oxides in the potential range, which is considered as the "real catalyst" of OER.^[14,15] For this reason, many researchers use reconstruction to obtain catalysts with high activity OER (Such as Small DOI: 10.1002/sml.202308613; Chem. Eng. J. 2022, 430:

132632; etc.). However, this continuous and disordered reconstruction process is fatal to stabilize the structure of the catalyst.^[16-18] In this study, an ultrathin amorphous MoO₃ chloride ion barrier layer was constructed on the CoO surface by ALD to achieve selective seawater oxidation and restricted dynamic surface self-reconfiguration. This "kill two birds with one stone" strategy avoids chlorine evolution reaction and catalyst phase separation due to too high polarization voltage. Therefore, we are different from the research object of literature.

The performance index in the literature (Adv. Energy Mater. 2023, 2301475; etc) is that it is stable for 600 h in 6M KOH+seawater electrolyte with current density of 500 mA cm⁻². It only lasts for 350 h at 1A cm⁻² requires a voltage of 2.25 V. In contrast, we reported: The flow electrolytic cell composed of MoO₃@CoO/CC and commercial Pt/C shows superior hydrogen production performance. The industrial current (1 A cm⁻²) only requires a voltage of 1.93 V, and the hydrogen evolution lasts for 500 h without attenuation, and the Faraday efficiency remains above 95%, showing high stability and selectivity. In addition, the hydrogen production rate is 419.4 mL cm⁻² h⁻¹, and the power consumption is only 4.62 kWh m⁻³ H₂, which is lower than the energy consumption of pure water electrolysis (~5 kWh m⁻³ H₂), showing superior application prospects. As shown in **Table R1**, the performance of the results far exceeds those reported in real seawater electrolysis (See Fig.7g and Table S7 of the manuscript for details).

To further clarify our innovation, we summarize as follows:

This paper mainly studies the following problems of anode in seawater electrolysis: First, CER is competitive to OER at anode that lowers the OER selectivity and forms toxic chlorine.^[19] Second, the strong binding energy between Cl⁻ and active sites of the electrocatalysts accelerates catalyst corrosion and leads to poor durability.^[20] Electrolysis at high current densities is crucial for practical applications, but the above problems become more serious than that at low current densities (<200 mA cm⁻²).^[21] As a result, the highest current densities delivered constantly by most of the seawater electrocatalysts reported so far remain below the industrial requirements of 500 mA cm⁻²,^[22] and it is rare that the electrocatalysts work stably for

over 200 h.^[23] Therefore, the main challenge of seawater electrolysis is how to prevent CER from selectively performing OER at anode.^[24] The other major challenge of seawater splitting is how to control the process of catalyst reconstruction to avoid phase separation caused by deep reconstruction, which becomes the key challenge to obtain high activity OER catalyst.^[25, 26]

For this reason, this paper focuses on the key scientific question of how to avoid deep catalyst reconfiguration and selective seawater oxidation. The designed MoO₃@CoO/CC catalysts were verified by a series of experiments to be capable of restricted dynamic surface self-reconstruction and selective OER during electrolytic seawater oxidation, and the related mechanisms were revealed in detail. Finally, we further evaluated the ability to design catalysts for the direct electrolysis of seawater by forming the anode and cathode of a flow electrolysis cell with MoO₃@CoO/CC and Pt/C, respectively. The assembled MoO₃@CoO/CC||Pt/C/CC cell achieves H₂ production rate of about 419.4 mL cm⁻² h⁻¹ for direct seawater electrolysis, and the corresponding power consumption is only 4.62 KWh/m³ H₂. These results exceed those reported in real seawater electrolysis. This discovery not only develops a robust and stable catalyst to utilize abundant seawater source for large-scale hydrogen production, but also encloses a restricted dynamic surface reconstruction mechanism for guiding the design of high-performance OER catalyst.

Table R1. Comparison of hydrogen production rate with literature. Comparison of hydrogen production rates in various seawater.

Material	Medium	H ₂ (mL cm ⁻² h ⁻¹)	Ref.
MoO ₃ @CoO/CC Pt/C/CC	Real seawater (25 °C)	419.4	This Work
Cr ₂ O ₃ -CoO _x Cr ₂ O ₃ -CoO _x	Alkalized Seawater (60 °C)	300	Nat. Energy , 8 , 264 (2023)
Ni-NiO-Cr ₂ O ₃ Ni-NiO-Cr ₂ O ₃	1 M KOH + 0.5 M NaCl	200	PNAS , 116 , 6624 (2019)

S-(Ni,Fe)OOH NiMoN	1 M KOH + seawater	48	Energy Environ. Sci. 13 , 3439 (2020)
Ir-C Pt-C	Real Seawater	1.8	Adv. Energy Mater. 8 , 1801926 (2018)
Pt SS	Seawater	3.3	PNAS , 108 , 16176 (2011).
NiCoN Ni _x P NiCoN NiCoN Ni _x P NiCoN	Real Seawater	4.5	ACS Energy Lett. 5 , 2681 (2020)
Ni _x B/B ₄ C/B-C _{PR} /NF-B5P1 Pt/C/NF	Alkaline seawater	1.0	ChemSusChem 14 , 5499-5507 (2021)
Ni-doped FeOOH Pt/C	Alkaline seawater	1.2	J. Mater. Chem. A 9 , 9586-9592 (2021).
NiCo@C/MXene/CF Pt/C	Seawater	255.1	Nat. Commun. 12 , 4182 (2021).
Ru-CoO _x /NF Ru-CoO _x /NF	Seawater	19.8	Small 17 , e2102777 (2021).

References:

- [1] Guo, J. *et al.*, Direct seawater electrolysis by adjusting the local reaction environment of a catalyst. *Nat. Energy* **8**, 264–272 (2023).
- [2] He, W. J. *et al.*, Materials Design and System Innovation for Direct and Indirect Seawater Electrolysis. *ACS Nano* **17**, 22227-22239 (2023).
- [3] Xie, H. P. *et al.*, A membrane-based seawater electrolyser for hydrogen generations, *Nature* **612**, 673-678 (2022).
- [4] Luo, Y. *et al.* Morphology and surface chemistry engineering toward pH-universal catalysts for hydrogen evolution at high current density. *Nat. Commun.* **10**, 269 (2019).
- [5] Wu, B. *et al.* A unique NiOOH@FeOOH heteroarchitecture for enhanced oxygen evolution in

- saline water. *Adv. Mater.* **34**, 2108619 (2022).
- [6] Yu, L. *et al.*, Ultrafast room-temperature synthesis of porous S-doped Ni/Fe (oxy)hydroxide electrodes for oxygen evolution catalysis in seawater splitting. *Energy Environ. Sci.* **13**, 3439-3446 (2020).
- [7] Yu, L. *et al.*, High-performance seawater oxidation by a homogeneous multimetallic layered double hydroxide electrocatalyst. *PNSA* **119**, e2202382119 (2022).
- [8] Xu, W. W. *et al.*, Ag Nanoparticle-induced Surface Chloride Immobilization Strategy enables Stable Seawater Electrolysis. *Adv. Mater.* **35** 2306062 (2023).
- [9] Kang, X. *et al.*, A corrosion-resistant RuMoNi catalyst for efficient and long-lasting seawater oxidation and anion exchange membrane electrolyzer. *Nat. Commun.* **14**, 3607 (2023).
- [10] Kuang, Y. *et al.*, Solar-driven, highly sustained splitting of seawater into hydrogen and oxygen fuels. *PNAS.* **116**, 6624-6629 (2019).
- [11] Liu W. X. *et al.*, Ferrum-molybdenum dual incorporated cobalt oxides as efficient bifunctional anti-corrosion electrocatalyst for seawater splitting. *Appl. Catal. B-Environ.* **328**, 122488 (2023).
- [12] Huang, Y. *et al.*, Plasma-induced Mo-doped Co₃O₄ with enriched oxygen vacancies for electrocatalytic oxygen evolution in water splitting. *Carbon Energy*, **5**, 279 (2022).
- [13] Ye, S. H. *et al.*, Deeply self-reconstructing CoFe(H₃O)(PO₄)₂ to low-crystalline Fe_{0.5}Co_{0.5}OOH with Fe³⁺-O-Fe³⁺ motifs for oxygen evolution reaction. *Appl. Catal. B: Environ.* **304**, 120986 (2022).
- [14] Wang, K. T. *et al.*, Mn Doping and P Vacancy Induced Fast Phase Reconstruction of FeP for Enhanced Electrocatalytic Oxygen Evolution Reaction in Alkaline Seawater. *Small* **20**, 2308613 (2023).
- [15] Chen Y. X. *et al.*, Utilizing tannic acid and polypyrrole to induce reconstruction to optimize the activity of MOF-derived electrocatalyst for water oxidation in seawater. *Chem. Eng. J.* **430**, 132632 (2022).
- [16] Minguzzi, A., How to improve the lifetime of an electrocatalyst. *Nat. Catal.* **3**, 687-689 (2020).
- [17] Wang, W. *et al.*, Structural Reconstruction of Catalysts in Electroreduction Reaction: Identifying, Understanding, and Manipulating. *Adv. Mater.* **34**, 2110699 (2022).
- [18] Kuai, C. *et al.*, Phase segregation reversibility in mixed-metal hydroxide water oxidation catalysts. *Nat. Catal.* **3**, 743-753 (2020).
- [19] Xie, H. P. *et al.*, A membrane-based seawater electrolyser for hydrogen generations, *Nature* **612**, 673-678 (2022).
- [20] Shi, L. *et al.*, Using reverse osmosis membranes to control ion transport during water electrolysis. *Energy Environ. Sci.* **13**, 3138-3148 (2020).
- [21] Luo, Y. *et al.* Morphology and surface chemistry engineering toward pH-universal catalysts for hydrogen evolution at high current density. *Nat. Commun.* **10**, 269 (2019).
- [22] Vos, J. G. *et al.*, MnO_x/IrO_x as selective oxygen evolution electrocatalyst in acidic chloride solution. *J. Am. Chem. Soc.* **140**, 10270-10281 (2018).
- [23] Wu, B. *et al.* A unique NiOOH@FeOOH heteroarchitecture for enhanced oxygen evolution in saline water. *Adv. Mater.* **34**, 2108619 (2022).
- [24] Kang, X. *et al.*, A corrosion-resistant RuMoNi catalyst for efficient and long-lasting seawater oxidation and anion exchange membrane electrolyzer. *Nat. Commun.* **14**, 3607 (2023).

- [25] Wang, W. et al., Structural Reconstruction of Catalysts in Electroreduction Reaction: Identifying, Understanding, and Manipulating. *Adv. Mater.* **34**, 2110699 (2022).
- [26] Karlsson, R. K. B. & Cornell, A., Selectivity between Oxygen and Chlorine Evolution in the Chlor-Alkali and Chlorate Processes. *Chem. Rev.* **116**, 2982-3028 (2016).